# Phagosomal signalling of the C-type lectin receptor Dectin-1 is terminated by intramembrane proteolysis

Torben Mentrup [1], Anna Yamina Stumpff-Niggemann[2], Nadja Leinung[1], Christine Schlosser[3], Katja Schubert[4], Rebekka Wehner[5,6,7], Antje Tunger[5,6], Valentin Schatz[8], Patrick Neubert[8], Ann-Christine Gradtke[1], Janina Wolf[2], Stefan Rose-John [2], Paul Saftig [2], Alexander Dalpke[9], Jonathan Jantsch[8], Marc Schmitz[5,6,7], Regina Fluhrer [3], Ilse D. Jacobsen[4,10] & Bernd Schröder [1✉]

Sensing of pathogens by pattern recognition receptors (PRR) is critical to initiate protective host defence reactions. However, activation of the immune system has to be carefully titrated to avoid tissue damage necessitating mechanisms to control and terminate PRR signalling. Dectin-1 is a PRR for fungal β-glucans on immune cells that is rapidly internalised after ligand-binding. Here, we demonstrate that pathogen recognition by the Dectin-1a isoform results in the formation of a stable receptor fragment devoid of the ligand binding domain. This fragment persists in phagosomal membranes and contributes to signal transduction which is terminated by the intramembrane proteases Signal Peptide Peptidase-like (SPPL) 2a and 2b. Consequently, immune cells lacking SPPL2b demonstrate increased anti-fungal ROS production, killing capacity and cytokine responses. The identified mechanism allows to uncouple the PRR signalling response from delivery of the pathogen to degradative compartments and identifies intramembrane proteases as part of a regulatory circuit to control anti-fungal immune responses.

[1] Institute of Physiological Chemistry, Faculty of Medicine Carl Gustav Carus, Technische Universität Dresden, Dresden, Germany. [2] Biochemical Institute, Christian-Albrechts-University of Kiel, Kiel, Germany. [3] Biochemistry and Molecular Biology, Institute of Theoretical Medicine, Medical Faculty, University of Augsburg, Augsburg, Germany. [4] Research Group Microbial Immunology, Leibniz Institute for Natural Product Research and Infection Biology, Hans Knöll Institute, Jena, Germany. [5] Institute of Immunology, Faculty of Medicine Carl Gustav Carus, Technische Universität Dresden, Dresden, Germany. [6] National Center for Tumor Diseases (NCT), Partner Site Dresden, Dresden, Germany. [7] German Cancer Consortium (DKTK), Partner Site Dresden, Dresden, and German Cancer Research Center (DKFZ), Heidelberg, Germany. [8] Institute of Clinical Microbiology and Hygiene, University Hospital of Regensburg and University of Regensburg, Regensburg, Germany. [9] Institute of Medical Microbiology and Virology, University Hospital Carl Gustav Carus, Medical Faculty, Technische Universität Dresden, Dresden, Germany. [10] Institute of Microbiology, Friedrich Schiller University, Jena, Germany. ✉email: bernd.schroeder@tu-dresden.de

Pattern-recognition receptors (PRRs) represent a key element of innate immunity initiating immediate responses to pathogens and thereby pave the way for adaptive immunity[1,2]. PRR activation triggers cytokine production, but also upregulates phagocytic, microbicidal and antigen-presentation capacity. Since excessive activation of immune cells can lead to tissue damage, this process has to be tightly controlled[3]. Therefore, cellular mechanisms to terminate PRR-triggered signalling after pathogen encounter are essential.

C-type lectin receptors (CLRs) represent a large group of membrane-bound PRRs for recognition of extracellular pathogens that share a carbohydrate-binding C-type lectin domain[4,5]. One of the best-characterised CLRs is Dectin-1 which recognises β-glucan structures that are particularly abundant in fungal cell walls[6]. Dectin-1-deficient mice show a strongly enhanced susceptibility to fungal infections documenting the pathophysiological relevance of this receptor and the associated pathways[7–9]. Furthermore, in humans a Y238X Dectin-1 polymorphism is associated with increased susceptibility to invasive aspergillosis in high-risk patients[10]. In both mouse and man, Dectin-1 is expressed in two major splice isoforms, Dectin-1a and Dectin-1b, which differ in one exon[11,12]. At the protein level, this corresponds to a membrane-proximal stalk domain in the extracellular part of the protein, that is part of the Dectin-1a isoform, but absent in Dectin-1b. Different mouse strains vary significantly in the expression of the two isoforms[11]. Nevertheless, very little is known about potential functional differences between these Dectin-1 variants.

Dectin-1 is a type II transmembrane protein exhibiting a hemi-immunoreceptor tyrosine-based activation motif (hemITAM) in its short N-terminal cytoplasmic tail. Upon ligand binding and receptor dimerisation, this motif becomes phosphorylated and initiates signal transduction via the kinase Syk[13] and the adaptor protein CARD9[14], leading to the activation of the MAP kinase[15] and NFκB[16] pathway. This triggers the secretion of pro-inflammatory cytokines and production of reactive oxygen species (ROS)[17]. Furthermore, Dectin-1 mediates phagocytosis and pathogen clearance[18].

Internalisation of Dectin-1 and delivery to lysosomes has been linked with a downregulation of its signalling response[19]. The canonical pathway for membrane protein degradation within the endocytic system involves sorting into intraluminal vesicles of multivesicular bodies (MVBs). This is usually mediated by the endosomal transport complexes required for transport (ESCRT) machinery and triggered by ubiquitination[20]. Recently, the E3 ubiquitin ligase Cbl-b was shown to ubiquitinate Dectin-1 thereby promoting its degradation[21–23]. Interference with Cbl-b activity delayed Dectin-1 degradation in a mouse strain that primarily expresses Dectin-1b, and by this means enhanced Dectin-1-induced production of cytokines and ROS.

Selected single-spanning transmembrane proteins can bypass the sorting into intraluminal vesicles for being degraded. Their turnover is facilitated by intramembrane proteases either directly at the plasma membrane or within the limiting membrane of endosomes/lysosomes[24–26]. Different families of intramembrane proteases can be distinguished based on their catalytic mechanism. They all share the capability to hydrolyse peptide bonds within the hydrophobic environment of the phospholipid bilayer[27]. The Signal peptide peptidase (SPP)/SPP-like (SPPL) family of GxGD-type aspartyl intramembrane proteases shows a specificity for substrate proteins with a type II membrane topology[28,29]. The two closely related family members SPPL2a and SPPL2b are predominantly localised in lysosomes/late endosomes and at the plasma membrane, respectively[30,31]. In cell-based overexpression assays, substrate spectra of SPPL2a and SPPL2b overlap extensively indicating similar catalytic

properties[32]. To date, two substrates have been validated in vivo: the CD74 protein[25,33,34], the invariant chain of the MHCII complex, and the CLR LOX-1, which is activated by oxidised LDL (oxLDL)[26]. Whereas under endogenous conditions intramembrane proteolysis of CD74 in vivo solely depends on SPPL2a, both proteases mediate the cleavage of LOX-1. In SPPL2a-deficient mice, accumulation of uncleaved CD74 N-terminal fragments (NTFs) leads to a distinct immunological phenotype characterised by an impairment of splenic B cell maturation and a significant depletion of different dendritic cell subsets[25,33,34]. Similarly, loss of SPPL2a in humans leads to a loss of conventional dendritic cells, in particular the cDC2 population, which results in a Mendelian Susceptibility to Mycobacterial Disease (MSMD)[35]. LOX-1 NTFs enhance the development of atherosclerosis in SPPL2a/b double-deficient mice[26].

In this study, we systematically compared the degradation pathways of the two major Dectin-1 isoforms. We demonstrate that Dectin-1a continues to signal from phagosomes based on a retention of the Dectin-1a NTF in the phagosomal membrane. This biologically active Dectin-1a fragment depends on SPPL2 intramembrane proteases for clearance from the membrane and thus for terminating its signalling function. These findings identify a previously unrecognised signalling mode of Dectin-1, which is regulated by intramembrane proteolysis and highlights SPPL2 proteases as negative modulators of anti-fungal immune responses.

## Results
**Differential degradation of Dectin-1 isoforms.** Previous studies revealed that Dectin-1 degradation is important for termination of its signalling[21–23]. However, they did not distinguish between the two major Dectin-1 isoforms, Dectin-1a and -1b, differing in the presence of a membrane-proximal stalk domain (Fig. 1a). Humans express these in a ratio of approximately 1:1 as observed in PBMCs from several individuals (Fig. 1b). Since overexpressed human Dectin-1b as reported previously[36] did not reach the cell surface but was retained in the ER (Supplementary Fig. 1a), we wanted to clarify the localisation of the human Dectin-1 isoforms at the endogenous level. In human PBMCs, ligand-induced degradation of Dectin-1b was slightly less complete than that of Dectin-1a (Supplementary Fig. 1b). In line, a significant fraction of Dectin-1b was not accessible to a membrane-impermeable biotinylation reagent supporting an intracellular localisation, whereas the cellular Dectin-1a pool was quantitatively localised at the plasma membrane (Supplementary Fig. 1c). As both isoforms were present at the cell surface of PBMCs they can be assumed to have a functional role in humans. In contrast, commonly employed C57BL/6 (BL6) mice express nearly exclusively the 1b isoform (Fig. 1c). The isoform pattern in BALB/c mice reflects the situation in humans much better with a prominent presence of the 1a isoform (Fig. 1c). It is currently unclear how the stalk may alter the biochemical properties of the Dectin-1 protein. Most studies characterising Dectin-1 have been performed in BL6 mice including those analysing the impact of Cbl-b[21–23]. The prominent presence of Dectin-1a in humans and the relevance of Dectin-1 degradation for signalling termination prompted us to analyse this for the two isoforms in parallel.

In HEK293 cells stably overexpressing murine Dectin-1b, which is efficiently reaching the cell surface (Supplementary Fig. 1d), the receptor was degraded after applying the ligand depleted Zymosan (dZym) (Fig. 1d). When performing the same experiment with Dectin-1a, we noticed the generation of a stable N-terminal fragment (NTF) with an apparent molecular weight of ~18 kDa (Fig. 1e). Though this fragment was also detected in the absence of ligands, its abundance was strongly enhanced by

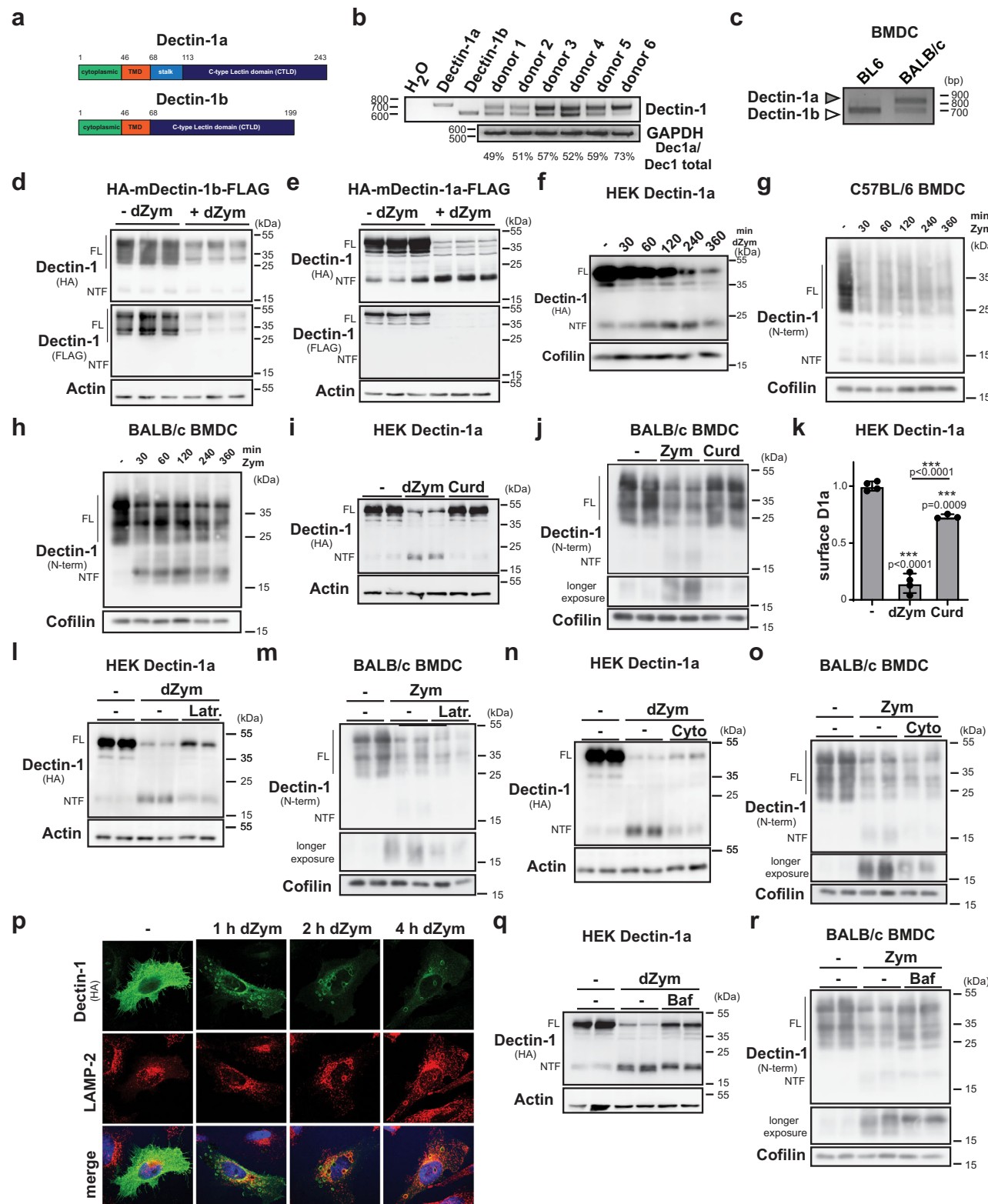

Dectin-1a stimulation. In comparison, only a very faint NTF band was seen in Dectin-1b expressing cells which was not modulated by Dectin-1 activation (Fig. 1d). In a time-course experiment (Fig. 1f), Dectin-1a NTF levels reached a maximum several hours after ligand application arguing for a significant stability of this fragment. These findings indicate that the degradative pathways of Dectin-1a and Dectin-1b differ. Presence of the stalk stabilises an NTF, which escapes from ESCRT-mediated degradation.

In order to analyse Dectin-1 processing at the endogenous level, we generated an antiserum against the Dectin-1 N-terminus, which specifically detects both isoforms upon over-expression (Supplementary Fig. 2a) as well as endogenously (Supplementary Fig. 2b). We compared Zymosan (Zym)-induced Dectin-1 degradation in bone marrow-derived dendritic cells (BMDCs) from BL6 (Fig. 1g) and BALB/c mice (Fig. 1h), which express predominantly the Dectin-1b or Dectin-1a isoforms,

**Fig. 1 Differential degradation of Dectin-1 isoforms. a** Scheme of murine Dectin-1 proteins. **b** Expression of Dectin-1 isoforms in human PBMCs was analysed by RT-PCR. $N = 1$, $n = 6$. **c** cDNA generated from RNA isolated from BMDCs of either C57BL/6 (BL6) or BALB/c mice was subjected to RT-PCR using Dectin-1-specific primers. $N = 3$, $n = 3$. **d** HEK cells stably overexpressing HA-mDectin-1b-FLAG were treated for 6 h with 50 µg/ml depleted Zymosan (dZym) or left untreated and subjected to western blot analysis. Full-length (FL) Dectin-1 and the corresponding N-terminal fragment (NTF) are highlighted throughout the Figure. $N = 2$, $n = 6$. **e** The experiment described in d) was repeated with HEK cells stably transfected with HA-mDectin-1a-FLAG. $N = 2$, $n = 6$. **f** Dectin-1a expressing HEK cells were treated for 0, 30, 60, 120, 240 or 360 min with 50 µg/ml dZym and analysed by western blotting. $N = 3$, $n = 3$ BMDC from C57BL/6 (**g**) or BALB/c (**h**) mice were stimulated for the indicated time points with 100 µg/ml Zymosan (Zym) and analysed for processing of Dectin-1 by western blotting. $N = 1$, $n = 3$ in both cases. **i** Dectin-1a expressing HEK cells were treated for 6 h with 50 µg/ml dZym or 200 µg/ml Curdlan (Curd) and subjected to western blot analysis. $N = 3$, $n = 6$. **j** The experiment in **i** was repeated with BALB/c BMDCs treated with 100 µg/ml Zym instead of dZym. $N = 3$, $n = 6$. **k** Surface expression of Dectin-1a was analysed in the setup described in **i** by flow cytometry. Bars depict Mean ± SD $N = 2$, $n = 3$ (Curd), $n = 4$ (dZym). One-way ANOVA with Tukey's post hoc testing. Stably Dectin-1a expressing HEK cells (**l** $N = 3$, $n = 6$) or BALB/c BMDCs (**m** $N = 3$, $n = 6$) were treated for 30 min with 5 µM Latrunculin A (Latr.) prior to stimulation with 50 µg/ml dZym or 100 µg/ml Zym for 6 h and detection of Dectin-1 by western blotting. The experiment was repeated in the same cells but with pre-treatment with 1 µM Cytochalasin D (Cyto) (**n**, **o**). $N = 3$, $n = 6$ in both cases. **p** HeLa cells transfected with HA-mDectin-1a-FLAG were treated with 50 µg/ml dZym, fixed and subjected to immunofluorescence analysis. Scale bar, 10 µm. $N = 3$, $n = 3$ HEK cells overexpressing HA-mDectin-1a-FLAG (**q** $N = 3$, $n = 6$) or BALB/c BMDC (**r**, $N = 3$, $n = 6$) were pretreated for 30 min with 100 nM Bafilomycin A1 (Baf) and stimulated for 30 min with 50 µg/ml dZym or 100 µg/ml Zym prior to western blot analysis. ***$p \leq 0.001$.

respectively (Fig. 1c). In both cases, Dectin-1 full-length receptors were degraded. In BALB/c BMDCs, a Dectin-1 NTF was produced in a ligand-dependent way that was reliably detected even 6 h after stimulation. In contrast, BL6 BMDCs exhibited only minor amounts of a smaller constitutively present Dectin-1 NTF (Fig. 1g) that was not influenced by ligand binding to the full-length receptor. Thus, the ligand-dependent generation of a Dectin-1a NTF occurs also under endogenous conditions raising the question how it is produced, which we analysed in parallel in overexpressing HEK cells and at the endogenous level in BMDCs.

**The Dectin-1a NTF is generated upon phagocytosis.** Dectin-1a NTF generation was induced by stimulation of the full-length receptor with dZym, which induces internalisation of Dectin-1[19,37]. Therefore, we wondered if NTF generation and internalisation are coupled. Whereas efficient NTF generation was induced by dZym or Zym, this was not observed with Curdlan (Fig. 1i, j), which also binds to Dectin-1, but due to the large particle size cannot be efficiently internalised (Fig. 1k)[19]. Furthermore, Latrunculin A (Fig. 1l, m) and Cytochalasin D (Fig. 1n, o), which both have been shown to block Dectin-1 endocytosis[19,37], significantly prevented NTF production upon dZym treatment. Interestingly, the full length form of Dectin-1a was not (BMDC, Fig. 1m, o) or only partly (HEK, Fig. 1l, n) stabilised by these compounds. Nevertheless, these findings indicate that internalisation is a prerequisite for Dectin-1a NTF generation. We also considered that proteolytic pathways at the cell surface might lead to the release of the Dectin-1a ectodomain. In both employed systems, the broad spectrum metalloproteinase inhibitor Marimastat had no impact on NTF levels (Supplementary Fig. 2c, d). Therefore, for murine Dectin-1 we have no indication that ectodomain shedding has a major contribution to NTF production.

The dependence on internalisation suggested that endosomal and/or lysosomal proteases are involved in Dectin-1a processing. In agreement with previous studies[38–40], we observed a recruitment of Dectin-1 to phagosomes and a co-localisation of Dectin-1a with the lysosomal marker protein LAMP-2 in transfected HeLa cells after dZym application (Fig. 1p). As observed before[38], inhibition of endosomal/lysosomal acidification with Bafilomycin A1 (Baf A1) partially inhibited dZym/Zym-induced degradation of full-length Dectin-1a (Fig. 1q, r) and additionally led to the stabilisation of some C-terminally truncated degradation intermediates. Interestingly, the Dectin-1a NTF was still generated in presence of Baf A1, but at a slightly reduced level. Conspicuously, the apparent molecular weight of the detected NTF seemed to be

increased under these conditions. Altogether, this indicates that acidic pH-dependent endosomal proteases are involved in NTF generation. We also assessed the impact of the broad-spectrum serine and/or cysteine protease inhibitors AEBSF (Supplementary Fig. 2e, f), E-64d and leupeptin (Supplementary Fig. 2g, h), which influenced Dectin-1a degradation and NTF generation in different ways. AEBSF reduced the amount of generated NTF. In contrast and reminiscent of Baf A1, the apparent molecular weight of the NTF produced in presence of leupeptin and E-64d was slightly higher than in control cells suggesting that a distinct processing step is blocked by these inhibitors. Altogether, these findings indicate that several proteases from different classes are involved in ligand-induced Dectin-1a degradation, which are partly dependent on endosomal acidification.

**Dectin-1a continues to signal from phagosomes.** The hemI-TAM motif within the Dectin-1 cytosolic domain, which is critical for signal transduction of the receptor[41], is preserved in the NTF. Therefore, we considered a contribution of this fragment to the signalling response. We analysed tyrosine-phosphorylated proteins isolated by immunoprecipitation (IP) from Dectin-1a transduced and Zym-treated J774.E macrophages (Fig. 2a). The detected amount of phosphorylated Dectin-1a full-length protein was highest 10 min after ligand application and then rapidly decreased presumably reflecting in part dephosphorylation but mainly the described degradation. We observed ligand-induced tyrosine-phosphorylation of the NTF, which followed different kinetics than that of the full-length receptor with a later maximum at 30 min. We confirmed phosphorylation of the endogenous Dectin-1a NTF generated upon Zym treatment of BMDCs from BALB/c mice (Fig. 2b). The amount of phosphorylated NTF increased here up to the end of the experiment at 120 min again highlighting a delayed phosphorylation response if compared to the full-length receptor. It seems likely that the NTF in its phosphorylated state may be able to recruit signalling proteins like the kinase Syk similar to the full-length receptor. In agreement, we observed a co-localisation of Dectin-1a and Syk-GFP around dZym particles in transiently transfected HeLa cells (Fig. 2c) in line with previous reports[13,42]. We also assessed association with Syk by co-immunoprecipitation in HEK cells overexpressing both Dectin-1a and Syk (Fig. 2d) as well as at the endogenous level in BALB/c BMDCs (Fig. 2e). In both systems, Zym triggered an interaction of full-length Dectin-1a with Syk. Importantly, also the Dectin-1a NTF co-immunoprecipitated with Syk in a ligand-induced way. We repeated this experiment with BL6 BMDCs since these cells also exhibit a faint NTF.

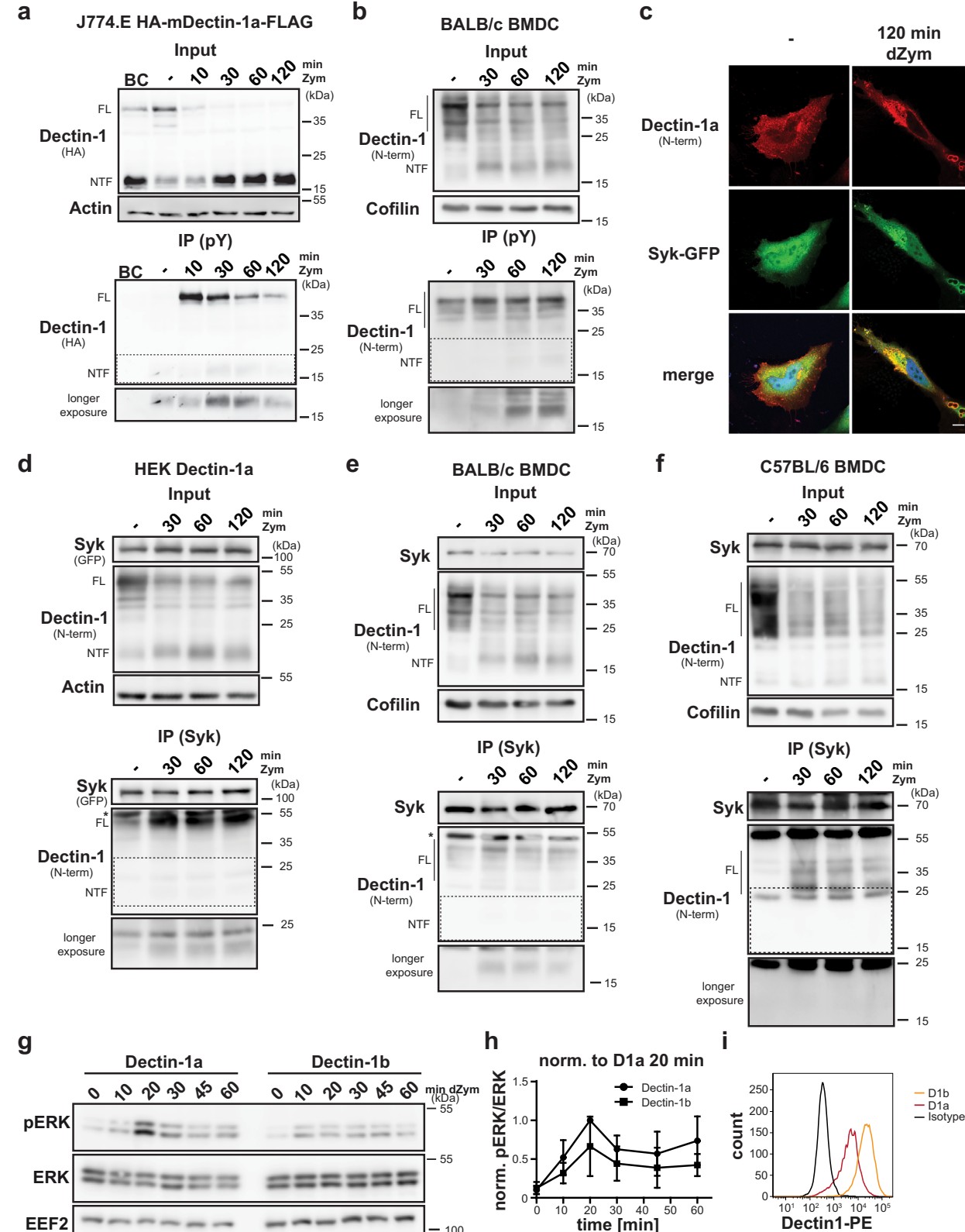

However, we failed to demonstrate co-immunoprecipitation of this fragment with Syk which is in agreement with its constitutive presence and lack of modulation by Dectin-1 activation excluding a relevance for Dectin-1 signalling (Fig. 2f). In contrast, the NTF-derived from Dectin-1a as in BALB/c BMDCs represents a previously unrecognised component of Dectin-1a signalling, which allows continued signalling activity after the internalised

pathogen has been delivered to a degradative and microbicidal compartment. This mechanism would suggest that based on the additional presence of a signalling-competent NTF the overall signalling capacity of Dectin-1a exceeds that of Dectin-1b. We tested this hypothesis using stable HEK cell lines expressing both isoforms. Treatment of these cells with dZym induces phosphorylation of the MAP kinases ERK1/2 (Fig. 2g) which is not

**Fig. 2 The Dectin-1a NTF persists in a signalling-competent state. a** J774.E cells stably expressing HA-mDectin-1a-FLAG were stimulated for the indicated time points using 100 µg/ml Zymosan (Zym). Lysates from cells treated for 20 min were employed as bead control (BC) where no precipitating antibody was added. After lysis, tyrosine-phosphorylated (pY) proteins were immune-precipitated (IP) using a phospho-Tyr antibody and protein G agarose beads. Lysates (input) and pull-down samples were analysed by western blotting. Full-length (FL) Dectin-1 and the corresponding N-terminal fragment (NTF) are highlighted throughout the Figure. N = 3, n = 3. **b** BMDC from BALB/c mice were subjected to the same experimental procedure described in **a**. N = 3, n = 3. **c** Co-localisation of Syk-GFP and HA-mDectin-1a-FLAG was validated in transiently transfected HeLa cells. Cells were treated for the indicated time points with 50 µg/ml depleted Zymosan (dZym) and subjected to immunofluorescence analysis using a Dectin-1-specific antibody. Scale bar, 10 µm. N = 3, n = 3. **d** Association of the Dectin-1a NTF with Syk was analysed in HEK cells stably expressing HA-mDectin-1a-FLAG transiently transfected with Syk-GFP. After lysis, Syk was precipitated with a specific antibody and protein G agarose beads and lysates as well as IP samples were subjected to western blot analysis employing the indicated antibodies. N = 3, n = 3. **e** Interaction of Syk and the Dectin-1a NTF was validated in BMDC of BALB/c wild type mice treated for 0, 30, 60 or 120 min with 100 µg/ml Zymosan. Syk was precipitated from lysates using a specific antibody and lysates and IP samples were analysed by western blotting using the indicated antibodies. N = 3, n = 3. **f** The experiment described in **e** was performed using BMDC isolated from C57BL/6 mice. N = 2, n = 4. **g** HEK cells stably overexpressing HA-mDectin-1a-FLAG or HA-mDectin-1b-FLAG were serum-starved for 1 h and subsequently stimulated for the indicated time points with 50 µg/ml dZym. Phosphorylation of ERK1/2 was finally evaluated by western blotting. N = 6; n = 6. **h** Quantification of **g**. N = 6; n = 6. Bars indicate Mean±SD. **i** Dectin-1 surface expression of the cell lines used in **h** was analysed by flow cytometry using a Dectin-1-specific antibody. N = 6, n = 6.

seen in vector-expressing cells confirming the specificity of the signalling response (Supplementary Fig. 3a). In this set-up, HEK cells overexpressing Dectin-1a demonstrated a stronger, but kinetically similar signalling response to dZym stimulation (Fig. 2g, h), even though surface levels of Dectin-1a were clearly lower than those of Dectin-1b in the corresponding cell line (Fig. 2i). These results support a stronger signalling capacity of Dectin-1a versus Dectin-1b, which would be in agreement with a relevant contribution of the phagosomal NTF-induced signalling.

**Dectin-1a NTF levels are controlled by SPPL2a and SPPL2b.** We wondered how the phagosomal Dectin-1a signalling may be regulated since the fragment seems to be segregated away from ESCRT-mediated degradation acting on the full-length receptor[21–23]. Intramembrane proteases are often involved in the sequential processing of single-spanning membrane proteins[27]. Based on the type II membrane topology and the subcellular localisation of Dectin-1a, proteases of the SPP/SPPL family represent candidates for processing the Dectin-1a NTF. Therefore, we assessed the impact of the broad spectrum SPP/SPPL inhibitor (Z-LL)$_2$-ketone (ZLL) on NTF levels in dZym-treated HEK cells expressing Dectin-1a and observed a significant stabilisation of this fragment (Fig. 3a). This effect was confirmed in J774.E cells stably expressing Dectin-1a (Supplementary Fig. 3b). We wanted to exclude that the failure to detect relevant amounts of a Dectin-1b NTF is based on an accelerated cleavage of this fragment by SPPL proteases. ZLL treatment slightly stabilised the Dectin-1b NTF. However, the amounts were still minor when compared to Dectin-1a (Supplementary Fig. 3c). We confirmed cleavage of the Dectin-1a NTF by SPPL proteases at the endogenous level by treating BMDCs from BALB/c mice with another SPP/SPPL inhibitor, inhibitor X (InX) (Fig. 3b). Based on the subcellular localisation, we considered SPPL2a and SPPL2b as candidate proteases. Therefore, we tested the general ability of these two proteases to cleave the Dectin-1a NTF. Upon co-expression in HEK cells, both SPPL2a and SPPL2b were able to cleave the Dectin-1a NTF to a similar degree (Fig. 3c) whereas inactive proteases with a D/A mutation were not. Additionally, in HeLa cells both proteases were recruited to Dectin-1a containing phagosomes (Fig. 3d, e) where the proteases and Dectin-1a co-localised in the limiting membrane of this compartment (Supplementary Fig. 3d, e). We also analysed processing of Dectin-1a in SPPL2a/b-deficient murine embryonic fibroblasts (MEFs). As compared to wild-type MEFs, we observed already basally, but in particular following dZym application, higher Dectin-1a NTF levels (Supplementary Fig. 3f). To narrow down if any of the two proteases has a leading role in cleaving the Dectin-1a NTF, we

performed siRNA-mediated knockdown in HEK293 cells stably overexpressing Dectin-1a (Fig. 3f and Supplementary Fig. 3g, h). Down-regulation of SPPL2b significantly stabilised the Dectin-1a NTF, whereas knockdown of SPPL2a in this cell system only had a very mild effect, although knockdown efficiency of both proteases was similar (Supplementary Fig. 3g, h). Upon re-analysis in SPPL2a-deficient HEK cells we observed an accumulation of the Dectin-1a NTF (Fig. 3g). This suggests that both SPPL2a and SPPL2b can contribute to the turnover of the Dectin-1 NTF.

We aimed to create an in vivo system for studying SPPL2a/b-mediated Dectin-1a cleavage in primary immune cells. Single and double knockout mice for SPPL2a/b have been reported[25,31], however, exclusively on a BL6 background, where Dectin-1a is not present in relevant amounts. Therefore, backcrossing to a Dectin-1a expressing background was required. Beyond Dectin-1, SPPL2a has another important function in immune cells, which is related to cleavage of the invariant chain CD74[25,33,34]. As CD74-associated phenotypes would be difficult to distinguish from effects of impaired Dectin-1 proteolysis, we decided to only backcross the SPPL2b knockout allele to a BALB/c background and characterise Dectin-1 proteolysis in BMDCs and bone marrow-derived macrophages (BMDM) from these mice.

After validation of the SPPL2b knockout by western blot analysis of BMDCs from backcrossed wild type and *SPPL2b*$^{−/−}$ mice (Fig. 3h), we started to analyse Dectin-1 proteolysis in these cells. The turnover of the ligand-induced NTF was significantly delayed by SPPL2b-deficiency (Fig. 3i). In contrast, SPPL2b deficiency did not increase the minor pool of constitutively present NTFs in BL6-derived cells (Fig. 3j). This confirmed that under endogenous conditions SPPL2b takes over part of the Dectin-1a NTF intramembrane cleavage and validates the system to study its functional consequences. However, treatment of SPPL2b-deficient BMDCs with inhibitor X significantly further enhanced the NTF stabilisation upon Zymosan-stimulation. This suggests a contribution of SPPL2a to NTF turnover in these cells (Fig. 3k). We verified these conclusions on SPPL2a/b-mediated turnover of Dectin-1a NTFs in BMDMs (Supplementary Fig. 3i–k).

**Intramembrane cleavage terminates Dectin-1a signalling.** Having revealed that the NTF contributes to Dectin-1a signalling and that its half-life depends on the intramembrane proteases SPPL2a/b, we wanted to study the role of the proteolytic cleavage for Dectin-1a-dependent signalling. In HEK cells stably over-expressing Dectin-1a, inhibition of SPPL proteases increased receptor-dependent MAP kinase activation (Fig. 4a). In agreement to our finding that Dectin-1b does not depend on SPPL2

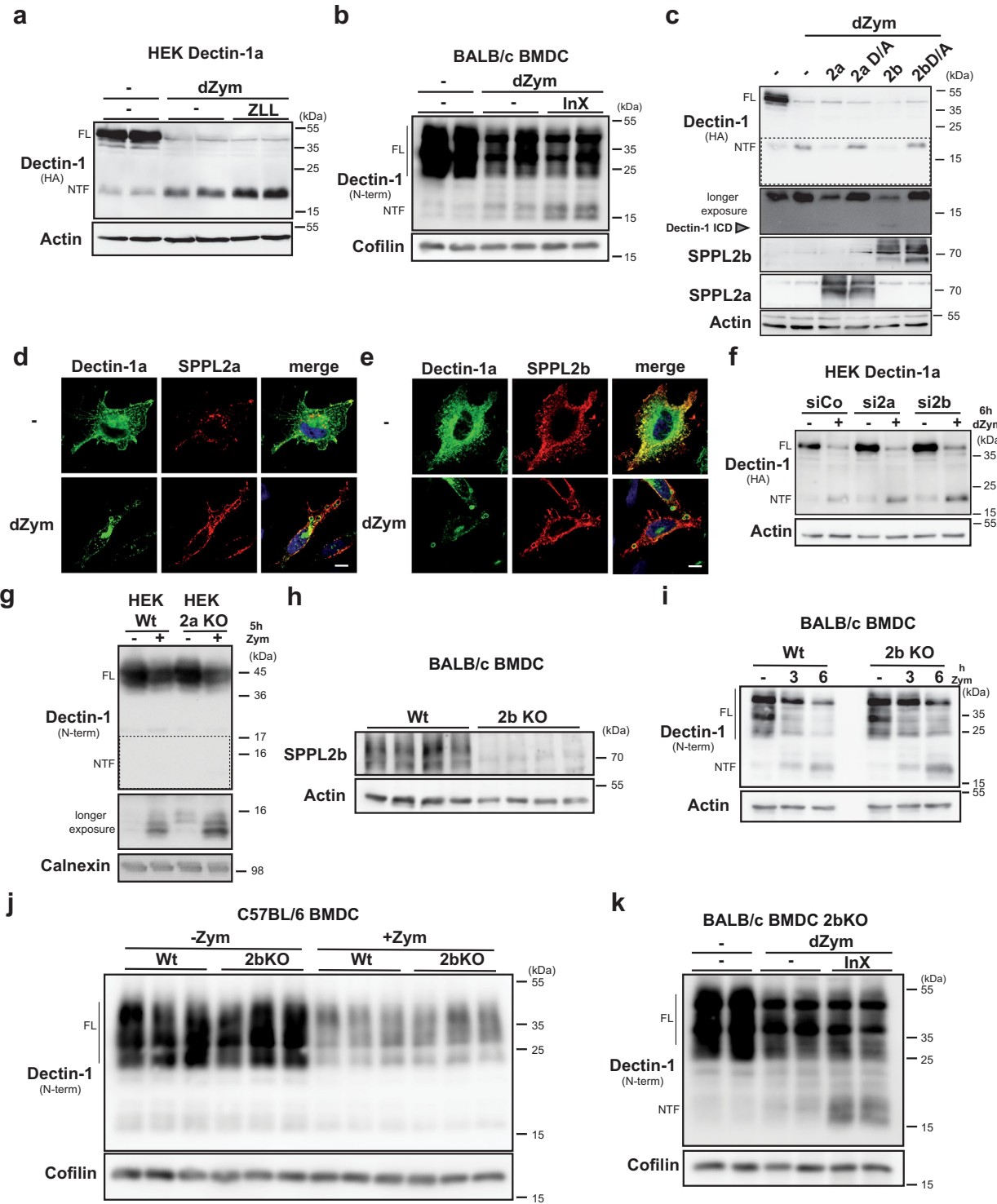

proteases for its degradation, ZLL did not significantly influence ERK1/2 phosphorylation in Dectin-1b expressing HEK cells (Fig. 4a). Knockdown of SPPL2b recapitulated the effects of the ZLL treatment (Fig. 4b). Again, siRNA against SPPL2a had, if at all, very subtle effects in line with its low impact on NTF proteolysis in this experimental system (Fig. 3f, Supplementary Fig. 3g, h). Altogether, these findings suggest that SPPL2 proteases terminate the Dectin-1a signalling response.

We wanted to test this hypothesis in BMDCs and BMDMs with endogenous expression of Dectin-1a. Inhibition of SPPL proteases with ZLL enhanced Dectin-1 triggered ERK activation

in BMDCs from BALB/c mice (Fig. 4c), but not from BL6 mice, which express mainly Dectin-1b (Fig. 4d). As a control, we determined the signalling response of BALB/c BMDC to the TLR4 ligand LPS, which was not modulated by ZLL (Supplementary Fig. 4a). We performed a knockdown of SPPL2b in BALB/c-derived BMDCs prior to stimulation with dZym (Fig. 4e and Supplementary Fig. 4b). Furthermore, we analysed Dectin-1 signalling in the BALB/c $SPPL2b^{-/-}$ BMDCs (Fig. 4f) and BMDMs (Supplementary Fig. 4c). Whereas Dectin-1-induced ERK signalling was enhanced in BALB/c $SPPL2b^{-/-}$ or knockdown cells, deficiency of the protease had no impact on LPS-

**Fig. 3 The Dectin-1a NTF is processed by SPPL2 proteases. a** HEK cells stably overexpressing HA-mDectin-1a-FLAG were treated for 30 min with 40 μM (Z-LL)$_2$-ketone (ZLL) prior to application of 50 μg/ml depleted Zymosan (dZym) for 6 h. Dectin-1a processing was visualised by western blotting using the indicated antibodies. Full-length (FL) Dectin-1 and the corresponding N-terminal fragment (NTF) are highlighted throughout the Figure. $N = 2$, $n = 4$.
**b** BALB/c BMDCs were pre-treated for 30 min with 1 μM inhibitor X (InX) and stimulated for 6 h with 50 μg/ml dZym prior to western blot detection of Dectin-1. $N = 4$, $n = 8$. **c** HEK cells were transiently transfected with HA-mDectin-1a-FLAG and the indicated murine SPPL2-encoding constructs. Where indicated, cells were treated for 6 h with 50 μg/ml dZym. Dectin-1 processing was finally analysed by western blotting. The grey arrow head marks the Dectin-1a intracellular domain (ICD). $N = 4$, $n = 4$. **d** HeLa cells were transiently transfected with HA-mDectin-1a and mSPPL2a-myc and treated for 1 h with 50 μg/ml dZym prior to immunofluorescence analysis employing anti-HA and anti-Myc as primary antibodies. Scale bar, 10 μm. **e** The experiment described in **d** was conducted with cells transfected with mSPPL2b-myc instead of mSPPL2a-myc. Scale bar, 10 μm. $N = 2$, $n = 2$. **f** Expression of SPPL2a or SPPL2b was down-regulated using 20 nM of a respective siRNA mix in HEK cells stably overexpressing HA-mDectin-1a-FLAG. After stimulation with 50 μg/ml dZym for 6 h, cells were analysed by western blotting. $N = 4$, $n = 4$. **g** Wild type (Wt) or SPPL2a-deficient (2a KO) T-Rex$^{TM}$-293 cells were transfected with HA-mDectin-1a-FLAG, stimulated for 4 h with 50 μg/ml Zym and subjected to western blotting. $N = 4$, $n = 4$. **h** Knockout of SPPL2b in BMDC was validated by western blotting. $N = 1$, $n = 4$. **i** BMDC from either Wt or SPPL2b knockout (2b KO) BALB/c mice were treated for 0, 3 or 6 h with 50 μg/ml dZym. After lysis, Dectin-1 processing was visualised by western blot analysis. $N = 5$, $n = 5$. **j** Wt or 2b KO C57BL/6 BMDC were treated with 100 μg/ml Zym and subsequently analysed by western blotting. $N = 2$, $n = 6$. **k** SPPL2b-deficient BMDCs from Balb/c mice were treated with 1 μM InX for 30 min prior to stimulation with 50 μg/ml dZym for 6 h. Dectin-1 levels were evaluated by western blotting $N = 3$, $n = 6$.

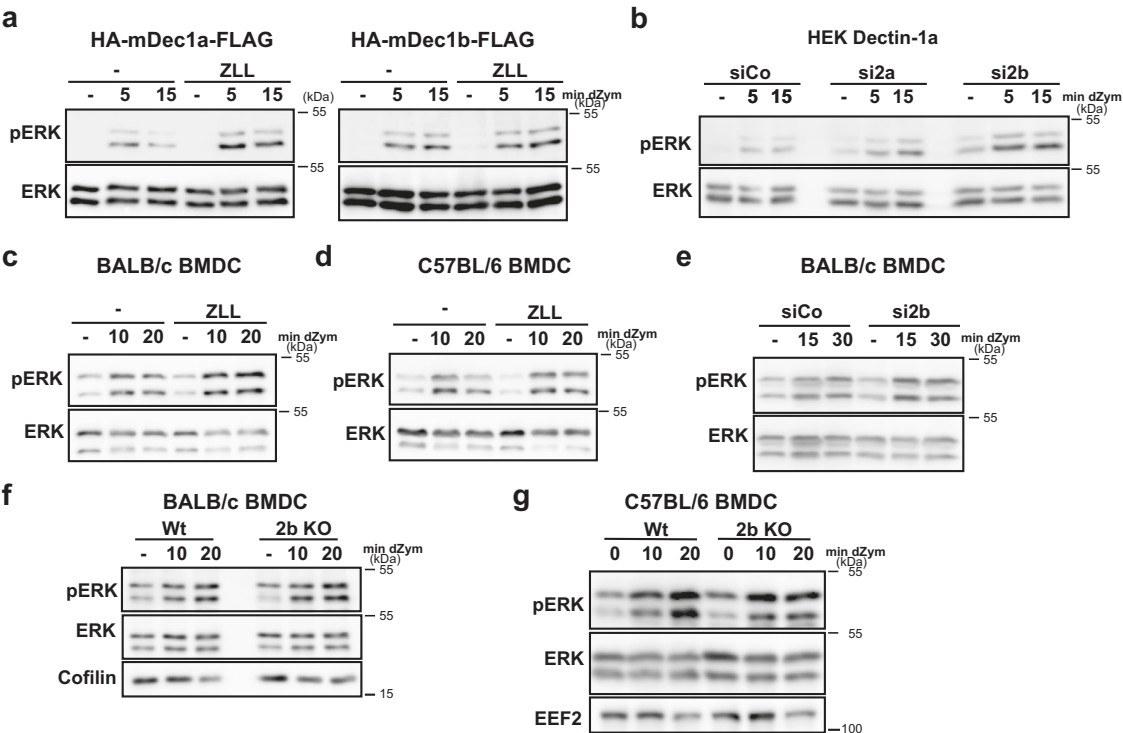

**Fig. 4 Intramembrane cleavage of the Dectin-1a NTF terminates signal transduction. a** HEK cells stably transfected with either HA-mDectin-1a-FLAG or HA-mDectin-1b-FLAG were serum-starved for 1 h and treated with 40 μM ZLL or the equivalent amount DMSO (−) for 30 min prior to stimulation with 50 μg/ml depleted Zymosan (dZym) for the indicated time points. Phosphorylation of ERK1/2 was evaluated using the indicated antibodies. Dectin-1a: $N = 5$, $n = 5$; Dectin-1b: $N = 4$, $n = 4$. **b** Expression of endogenous SPPL2a and SPPL2b was down-regulated by treatment with 50 nM of corresponding specific siRNAs in HEK cells stably overexpressing Dectin-1a. After serum starvation for 1 h, cells were stimulated with 50 μg/ml dZym and subjected to western blot analysis for ERK1/2 phosphorylation. $N = 3$, $n = 5$. **c** BALB/c BMDC were serum-starved for 1 h in the presence of either DMSO or 40 μM ZLL. Subsequently, cells were treated for the indicated time points with 50 μg/ml dZym. Levels of phosphorylated as well as total ERK1/2 were visualised by western blotting. $N = 2$, $n = 6$. **d** BMDC from C57BL/6 mice were treated as described in **c**. $N = 2$, $n = 4$. **e** BALB/c BMDC were treated with either control or SPPL2b-targeting siRNA. Subsequently, cells were subjected to analysis of signal transduction as described in previous subfigures. $N = 1$, $n = 3$. **f** Either wild type (Wt) or SPPL2b-deficient (2b KO) BALB/c BMDC were serum-starved for 1 h and subsequently treated for the indicated time points with 50 μg/ml dZym. Activation of ERK1/2 was determined by western blotting. $N = 2$, $n = 6$. **g** The same experiment was performed with mice with the same genotypes but on C57BL/6 background. $N = 2$, $n = 6$.

induced MAP kinase activation (Supplementary Fig. 4d) or on Dectin-1 signalling in BL6 cells (Fig. 4g and Supplementary Fig. 4e). Collectively, these experiments indicate that modulations of the Dectin-1a NTF turnover influence receptor-induced MAP kinase activation and that SPPL2 intramembrane proteases represent a novel mechanism of terminating Dectin-1 signalling.

**Cleavage by SPPL2b modulates Dectin-1a induced ROS production.** We wanted to assess if the enhanced MAP kinase responses upon SPPL2b inhibition or deficiency also translate into increased anti-fungal downstream responses. Therefore, we determined ROS production in BMDCs with genetic deficiency of SPPL2b (Fig. 5a, b). ROS production induced by dZym was

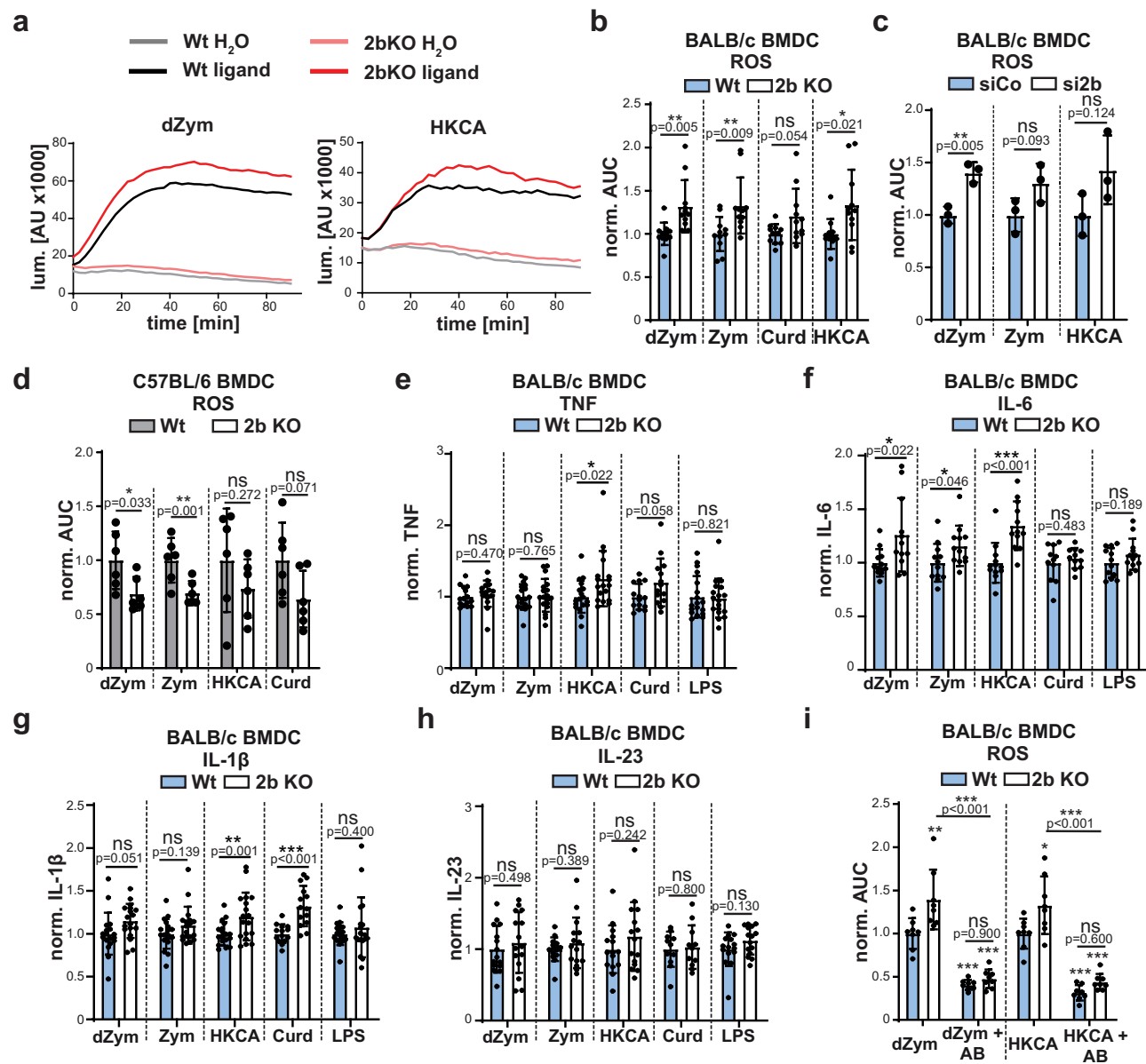

**Fig. 5 SPPL2b regulates ROS production in response to Dectin-1 ligands. a** BMDCs from BALB/c wild type (Wt) or SPPL2b knockout (2b KO) mice were treated with 50 μg/ml depleted Zymosan (dZym) or MOI 10 heat-killed *C. albicans* (HKCA). ROS formation was detected using the luminometric L-012 probe. The graphs depict a representative experiment. $N = 3$, $n = 11$. **b** Quantification of ROS formation from experiment shown in a) based on Area under the curve (AUC) calculation. Cells were stimulated with either 50 μg/ml dZym, 50 μg/ml Zym, MOI 10 HKCA or 200 μg/ml Curdlan. $N = 3$, $n = 11$. Two-tailed unpaired Student's *t* test. **c** ROS analysis was performed in wild type BALB/c BMDC transfected with either control or SPPL2b-targeting siRNA. $N = 1$, $n = 3$. Two-tailed unpaired Student's *t* test. **d** ROS formation was analysed in Wt or 2b KO C57BL/6 BMDC upon treatment with either 50 μg/ml dZym, 50 μg/ml Zym, MOI 10 HKCA or 200 μg/ml Curdlan. $N = 2$, $n = 6$. Two-tailed unpaired Student's *t* test. **e–h** Wild type or *SPPL2b$^{-/-}$* BMDC were treated for 24 h with either 50 μg/ml dZym, 50 μg/ml Zym, MOI 10 HKCA, 200 μg/ml Curdlan (Curd) or 500 ng/ml LPS as control. Finally, levels of TNF (**e**), IL-6 (**f**), IL-1β (**g**) or IL-23 (**h**) in the conditioned medium were measured by ELISA. TNF: Curd: $N = 4$, $n = 14$, dZym: $N = 4$, $n = 15$; others $N = 5$, $n = 18$. IL-6: Curd: $N = 3$, $n = 11$, rest $N = 4$, $n = 12$. IL-1β: Curd: $N = 4$, $n = 14$; rest $N = 5$, $n = 18$. IL-23: Curd: $N = 3$, $n = 11$, rest $N = 4$, $n = 15$. Two-tailed Unpaired Student's *t* test. **i** Prior to stimulation with either 50 μg/ml dZym or MOI 10 HKCA, wild type or SPPL2b-deficient BMDC were incubated with 10 μg/ml anti-Dectin-1 blocking antibody (AB) for 30 min. Subsequently, ROS production was determined. $N = 2$, $n = 8$. One-Way ANOVA with Tukey's *post hoc* test. p values from statistical testing for comparison of each condition to wild type control samples (from left to right) are: dZym: $p = 0.0031$, $p < 0.0001$, $p < 0.0001$; HKCA: $p = 0.9747$, $p = 0.2621$, $p > 0.0001$. All diagrams depict Mean values ± SD.

significantly augmented in these cells. Importantly, also more complex ligands like Zymosan (Zym) and heat-killed *C. albicans* (HKCA), which activate further PRRs in addition to Dectin-1, led to higher ROS levels in the knockout cells. In contrast, responses to the non-internalisable Dectin-1 ligand Curdlan were unaltered by the protease deficiency. Similar effects were observed in

BMDCs following knockdown of SPPL2b (Fig. 5c), where ROS induced by dZym, Zymosan and HKCA were increased. Furthermore, the augmented ROS generation was not restricted to BMDCs, but also present in SPPL2b-deficient BMDMs (Supplementary Fig. 5a). According to our model, loss of SPPL2b would not be expected to enhance ROS production in cells from BL6

mice, which was the case (Fig. 5d and Supplementary Fig. 5b). Unexpectedly, ROS levels were even significantly lower in the SPPL2b-deficient BL6 cells.

We also determined the impact of SPPL2b-deficiency on Dectin-1 induced cytokine responses in BALB/c BMDCs (Fig. 5e–h). Though the production of some cytokines after application of certain ligands was significantly enhanced, the amplitude was rather small in SPPL2b single-deficient cells. Interestingly, differences were mainly observed with the complex ligand HKCA, whereas cytokine production with dZym was essentially unaffected. Similar to ROS, also cytokine secretion of BL6-derived SPPL2b-deficient BMDCs was in many cases significantly lower than that of controls (Supplementary Fig. 5c–e), however, also following LPS, which points to some general SPPL2b-associated, but not Dectin-1-dependent impairment of cell activation. In BMDMs only secretion of TNF, but not of IL-1β and IL-6, could be reliably detected and only following stimulation with Zymosan (Supplementary Fig. 5f).

With regard to our model, we were puzzled that SPPL2b-deficiency in the Dectin-1a expressing background clearly enhanced ROS production (Fig. 5a, b and Supplementary Fig. 5a) whereas cytokine production induced by the same ligands was not or only very mildly modulated (Fig. 5e–h). Therefore, we assessed the Dectin-1-dependence of the analysed down-stream responses using a Dectin-1 blocking antibody (Fig. 5i). This reagent reduced the overall dZym- as well as HKCA-associated ROS response in BALB/c BMDCs from both wild type and SPPL2b-deficient mice confirming the requirement of Dectin-1. Importantly, it also attenuated the difference between cells of both genotypes thereby confirming that the observed responses in the latter are indeed caused by augmented Dectin-1 signalling. These findings were recapitulated in BMDMs (Supplementary Fig. 5g). However, the blocking antibody had no or extremely minor effects on secretion of TNF, IL-6 or IL-1β of BALB/c BMDCs activated by dZym and HKCA (Supplementary Fig. 5h–j). This indicates that in the employed experimental system these responses were largely independent of Dectin-1 activation and presumably mainly driven by other PRRs. In line with this, they were not modulated by SPPL2b deficiency.

As we could confirm that ligand-induced NTF generation and cleavage by SPPL2 proteases is conserved for human Dectin-1a (Supplementary Fig. 6a–c), we tested how SPPL2 inhibition affects production of ROS (Supplementary Fig. 6d, h) or pro-inflammatory cytokines (Supplementary Fig. 6e–g, i, j) by human PBMCs upon contact with Dectin-1 ligands (Supplementary Fig. 6d–g) or viable C. albicans (Supplementary Fig. 6h–j). Potential effects due to co-inhibition of γ-secretase by inhibitor X were controlled by DAPT, which also inhibits γ-secretase, but not SPPL2 proteases. Except a tendency of a slightly enhanced IL-6 secretion upon dZym stimulation following SPPL inhibition, no major effects supporting our hypothesis were observed. Unfortunately, this could not be confirmed upon treatment of PBMC with living C. albicans since we could not reliably detect IL-6 under these conditions. In contrast to the initial hypothesis, it rather seemed that ROS production induced by dZym and heat-killed C. albicans (HKCA) was diminished by the applied inhibitors (Supplementary Fig. 6d). A major concern with this approach was the broad activity of the utilised compounds including most SPP/SPPL proteases. SPPL2a-deficient antigen-presenting cells including dendritic cells (DCs), macrophages and B cells accumulate large amounts of a CD74 NTF, which disturbs intracellular trafficking and signal transduction[43]. Surface expression of Dectin-1 is reduced in a CD74-NTF-dependent manner in $SPPL2a^{-/-}$ BMDCs, which has a negative impact on Dectin-1 induced cytokine responses[44]. We observed that already a 1 h treatment led to a detectable CD74 NTF accumulation in human

PBMCs (Supplementary Fig. 7a) but also in in murine BMDMs (Supplementary Fig. 7b) and BMDCs (Supplementary Fig. 7c). Based on the compromised ROS production in the InX-treated PBMCs, we compared ROS generation in wild type and SPPL2a-deficient BMDCs from BL6 mice, where direct effects on Dectin-1 NTF proteolysis are not to be expected. Loss of SPPL2a reduced ROS production by Dectin-1 ligands by ~50% - presumably due to accumulation of the CD74 NTF (Supplementary Fig. 7d). In lack of a SPPL2b-specific inhibitor and due to the intrinsic difficulties to genetically modify primary human immune cells, the proposed concept cannot be tested in these cells. Therefore, we abandoned this approach and went back to the already introduced SPPL2b-deficient mice on BALB/c background. As SPPL2b is not involved in CD74 cleavage in vivo[31], confounding CD74 effects can be excluded in this system.

**SPPL2b modulates ROS and cytokine production induced by C. albicans.** An important question is whether the changes observed in SPPL2b-depleted BMDCs and BMDMs upon stimulation with artificial Dectin-1 ligands are also relevant in a more physiological context. Therefore, we co-cultured wild type and $SPPL2b^{-/-}$ BMDCs with C. albicans which induced generation of the Dectin-1 NTF. In the SPPL2b-deficient cells, NTF levels were significantly higher (Fig. 6a). In agreement with the results obtained with PRR model ligands, ROS production induced by C. albicans was augmented by loss of SPPL2b in BALB/c BMDCs (Fig. 6b, c), but not in BL6-derived cells (Fig. 6d). In BMDM this effect was not seen (Supplementary Fig. 8a, b). At all tested MOIs, $SPPL2b^{-/-}$ BMDCs from BALB/c mice secreted significantly more TNF (Fig. 6e) and IL-1β (Fig. 6f). In contrast, IL-6 (Fig. 6g) production was only increased at higher MOIs whereas IL-23 (Fig. 6h) levels were not changed in the protease-deficient cells. In BMDMs only TNF secretion was reliably detected, which was significantly increased in $SPPL2b^{-/-}$ cells at MOI 10 (Supplementary Fig. 8c). In contrast, SPPL2b-deficiency on a BL6 background was associated with significantly reduced production of TNF (Fig. 6i), IL-1β (Fig. 6j) and IL-6 (Fig. 6k) in BMDCs, whereas TNF release by BMDMs was not affected by loss of the protease (Supplementary Fig. 8d). Again, we analysed to which extent the cellular responses induced by C. albicans were dependent on Dectin-1. Blocking Dectin-1 significantly diminished ROS production in BMDCs and abolished the difference between wild type and $SPPL2b^{-/-}$ cells (Fig. 6l). In contrast, the ROS production in BMDMs as measured by the employed assay did not seem to be Dectin-1-dependent (Supplementary Fig. 8e). In case of cytokine secretion by BMDCs (Fig. 6m–o), the Dectin-1 antibody reduced or in the case of IL-6 abolished the increment seen in the SPPL2b knockout cells. However, the overall amplitude of the response in wild type cells was not affected suggesting that it is to a major part Dectin-1-independent. On the other hand, TNF secretion by BALB/c BMDMs in response to C. albicans could be quite effectively diminished by preventing Dectin-1 activation (Supplementary Fig. 8f).

**SPPL2a/b are part of a regulatory anti-fungal circuit.** These findings clearly indicate that SPPL2b deficiency limits and controls anti-fungal immune responses in BMDCs and BMDMs, however, with varying impact depending on the analysed readout and cell-type. As revealed by the blocking experiments, also the Dectin-1 dependence of the individual responses varied. Altogether, there seemed to be a good correlation between Dectin-1 dependence and regulation by SPPL2b supporting the proposed model on NTF-based Dectin-1 signalling and its termination by SPPL2-mediated intramembrane proteolysis (Fig. 7a).

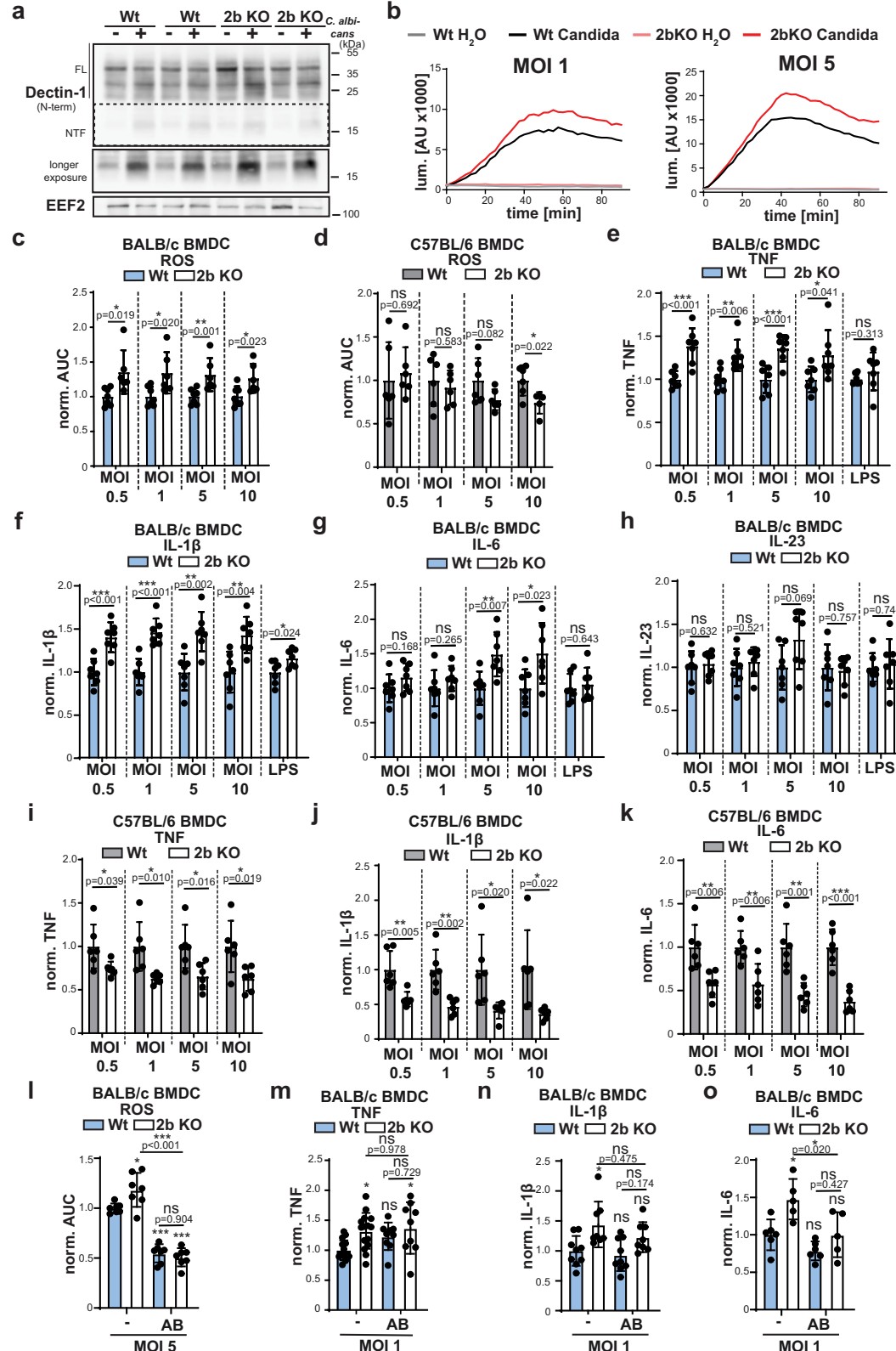

In general, other CLRs contribute to anti-fungal responses[4,5] and it cannot be excluded that also these are modulated by SPPL2b. Therefore, we asked if other CLRs, which have a documented role in sensing *C. albicans*, may undergo proteolytic processing. We analysed Mincle (Supplementary Fig. 9a) and Dectin-2 (Supplementary Fig. 9b) upon stable overexpression in J774.E cells as well as DC-SIGN (Supplementary Fig. 9c) upon

transient expression in HEK cells. We applied a broad range of artificial ligands described to interact with the respective receptors. Interestingly, in contrast to Dectin-1 none of the tested full-length receptors underwent significant ligand-induced degradation. However, in all cases we could observe low levels of constitutively present NTFs, which have not been described so far. In case of Mincle and DC-SIGN a mild increase in NTF levels

**Fig. 6 SPPL2b controls the response of BMDC to *Candida albicans*. a** Either wild type (Wt) or SPPL2b-deficient (2b KO) BALB/c BMDC were treated for 6 h with MOI 10 living *C. albicans* and analysed by western blotting for processing of Dectin-1. Full length (FL) Dectin-1 and the corresponding N-terminal fragment (NTF) are indicated. $N = 2$, $n = 4$. **b** Either Wt or 2b KO BMDC were stimulated for 90 min with viable *C. albicans* in the indicated MOIs. ROS formation was measured using the luminescent probe L-012. ROS curves of cells from representative individual animals are shown. $N = 2$, $n = 6$ (Wt) or 7 (2b KO). **c** Wt or 2b KO BALB/c BMDC were treated with living *C. albicans* in the indicated MOIs and analysed for ROS formation. ROS production was quantified based on area under the curve (AUC) calculation. $N = 2$, $n = 6$ (Wt) or 7 (2b KO). Two-tailed unpaired Student's *t* test. **d** The same experiment was conducted with BMDC from Wt or 2b KO mice on C57BL/6 background. $N = 2$, $n = 5$ (2b KO MOI 10) or $n = 6$ (rest). Two-tailed unpaired Student's *t* test. **e–h** Wt or 2b KO BMDC were treated for 24 h with viable *C. albicans* in the indicated MOIs or 500 ng/ml LPS and assayed for production of TNF (**e**), IL-1β (**f**), IL-6 (**g**) or IL-23 (**h**) by ELISAs. $N = 2$, $n = 7$ in all cases. Two-tailed unpaired Student's *t* test. Levels of TNF (**i**), IL-1β (**j**) and IL-6 (**k**) were also assessed in supernatants of equivalently treated BMDC with the same genotypes but on C57BL/6 background. $N = 2$, $n = 6$ in all cases. Two-tailed unpaired Student's *t* test. **l** Wt or 2b KO BMDC (Balb/c background) were pretreated with 10 μg/ml Dectin-1 blocking antibody (AB) and then stimulated with living *C. albicans* yeasts (MOI 1). ROS formation was analysed using the luminometric L-012 probe. $N = 3$, $n = 7$. One-Way ANOVA with Tukey's post hoc testing. p values from statistical testing for comparison of each condition to wild type control samples (from left to right) are: $p = 0,0214$, $p < 0.0001$, $p < 0.0001$. **m–o** BMDC from either Wt or 2b KO BALB/c mice incubated with anti-Dectin-1 (10 μg/ml) for 30 min prior to stimulation with living *C. albicans* yeast for 24 h. Levels of TNF (**m**), IL-1β (**n**) and IL-6 (**o**) in the media were measured by ELISA TNF: no blocking: $N = 5$, $n = 15$, blocking $N = 3$, $n = 9$. IL-1β: Wt: $N = 3$, $n = 9$, 2b KO: $N = 3$, $n = 8$. IL-6: Wt: $N = 3$, $n = 9$, 2b KO $N = 3$, $n = 8$. One-way ANOVA with Tukey's post hoc test. p values from statistical testing for comparison of each condition to wild type control samples (from left to right) are: TNF: $p = 0.0163$, $p = 0,2253$, $p = 0,017$; IL-1β: $p = 0.0195$, $p = 0.9592$, $p = 0,3809$; IL-6: $p = 0.0142$, $p = 0.3894$, $p > 0.999$. All diagrams depict Mean values ± SD. ns not significant; *$p \leq 0.05$; **$p \leq 0.01$; ***$p \leq 0.001$.

could be observed upon incubation with either HKMT, Zym or HKCA. Importantly, ZLL application moderately increased NTF levels at all tested conditions for these three receptors indicating that SPPL2 proteases might be involved in their clearance. This could indicate that SPPL2 proteases also modulate functions of other anti-fungal CLRs.

Altogether, the observed increment of ROS and/or cytokine production in absence of SPPL2b was rather small raising the question regarding the pathophysiological relevance of these effects. We compared the killing capacity of wild type and SPPL2b-deficient BMDCs (Fig. 7b) and BMDMs (Fig. 7c) upon co-culture with *C. albicans*. Phagocytosis of yeast cells was similar by the immune cells of both genotypes. However, the SPPL2b-deficient cells were significantly more effective in pathogen killing. This reveals, that the regulation of ROS production exerted by SPPL2b occurs in a relevant range.

Based on the newly discovered signalling terminating function of SPPL2a/b for Dectin-1a, we hypothesised that activation of this receptor or other PRRs may affect protease levels thereby modulating the negative regulation of signalling. Therefore, we compared SPPL2a/b protein levels in activated versus non-activated BMDCs. Both proteases were significantly upregulated upon stimulation of these immune cells by different CLR and TLR ligands (Fig. 7d). This reveals a new regulatory circuit in order to limit pathogen-induced immune cell activation (Fig. 7a) and in particular to fine-tune ROS production.

## Discussion

As one of the first lines of immune defence, PRRs like Dectin-1 sense pathogens and activate immune cells at a low threshold. However, once the system is primed, negative regulation is crucial to avoid over-activation of the immune system[3]. This could lead to tissue injury, particularly by generation of ROS[45]. Therefore, mechanisms to terminate PRR signalling are essential as characterised for TLRs[46,47]. In some cases, disturbance of these regulatory pathways has been linked to autoimmunity[46]. For a long time, very little was known about how Dectin-1 signalling is terminated. In principle, tyrosine phosphatases could be involved by dephosphorylating the Dectin-1 hemITAM. However, beyond the plasma membrane tyrosine phosphatases CD45 and CD148, which limit Dectin-1 activation within the phagocytic synapse[48], a role of such enzymes in controlling signalling strength or duration is not known yet. In this context, the finding that internalisation leading to receptor degradation terminates Dectin-

1 signalling was an important discovery[19,21–23]. In the proposed model, receptor degradation follows the MVB pathway as it has been extensively characterised for the EGF receptor[49]. According to our data, this model is incomplete since it has only been validated for the stalkless Dectin-1b isoform. In mice, this reflects the situation in BL6 mice which nearly exclusively express this isoform and which have been analysed in all respective studies[21–23]. Notably, these findings cannot be easily transferred to other mouse strains with different isoform expression patterns. More importantly, they do not reflect the human situation. Our data demonstrate that the degradative pathway of Dectin-1 differs significantly dependent on the presence of the stalk domain and reveal a novel mode of Dectin-1 signalling. The major functional consequence of the differential degradation is the prolonged persistence of the signalling-activating domain of Dectin-1a in the cytosol when compared to Dectin-1b. In the latter case, prior to the actual receptor degradation sorting into intraluminal vesicles immediately abrogates signalling by blocking access to cytosolic kinases. In contrast, the Dectin-1a-derived NTF persists in the limiting phagosomal membrane and continues to signal as documented by its phosphorylation and association with Syk. It seems possible, that such a mechanism increases the signalling capacity of the receptor and possibly reduces the signalling threshold, which may be advantageous for the gatekeeper function of a PRR. Though direct comparisons of the two isoforms are difficult, our results in Fig. 2g, h seem to support this view. Beyond degradation, it was reported before, that the ligand binding properties of the two murine isoforms may differ – however mainly at low temperatures, which may be of limited physiological relevance[11].

In the early phase of an infection, where maximal activation of the system with high sensitivity is required, it makes sense to uncouple receptor degradation and signalling termination. The Dectin-1a signalling mode, which we have characterised here, allows that pathogens bound by Dectin-1 can be delivered to degradative and microbicidal compartments without delay. At the same time activation of signalling can continue from the limiting membrane. Continued signalling from endosomes after internalisation has also been reported for TLR4, where internalisation leads to a switch of activated pathways[50]. In the case of Dectin-1a, we have no evidence that the NTF-triggered signalling is of a different quality than that of the full-length receptor.

An interesting question is how the Dectin-1 NTF is retained in the phagosomal membrane. We cannot exclude that a minor

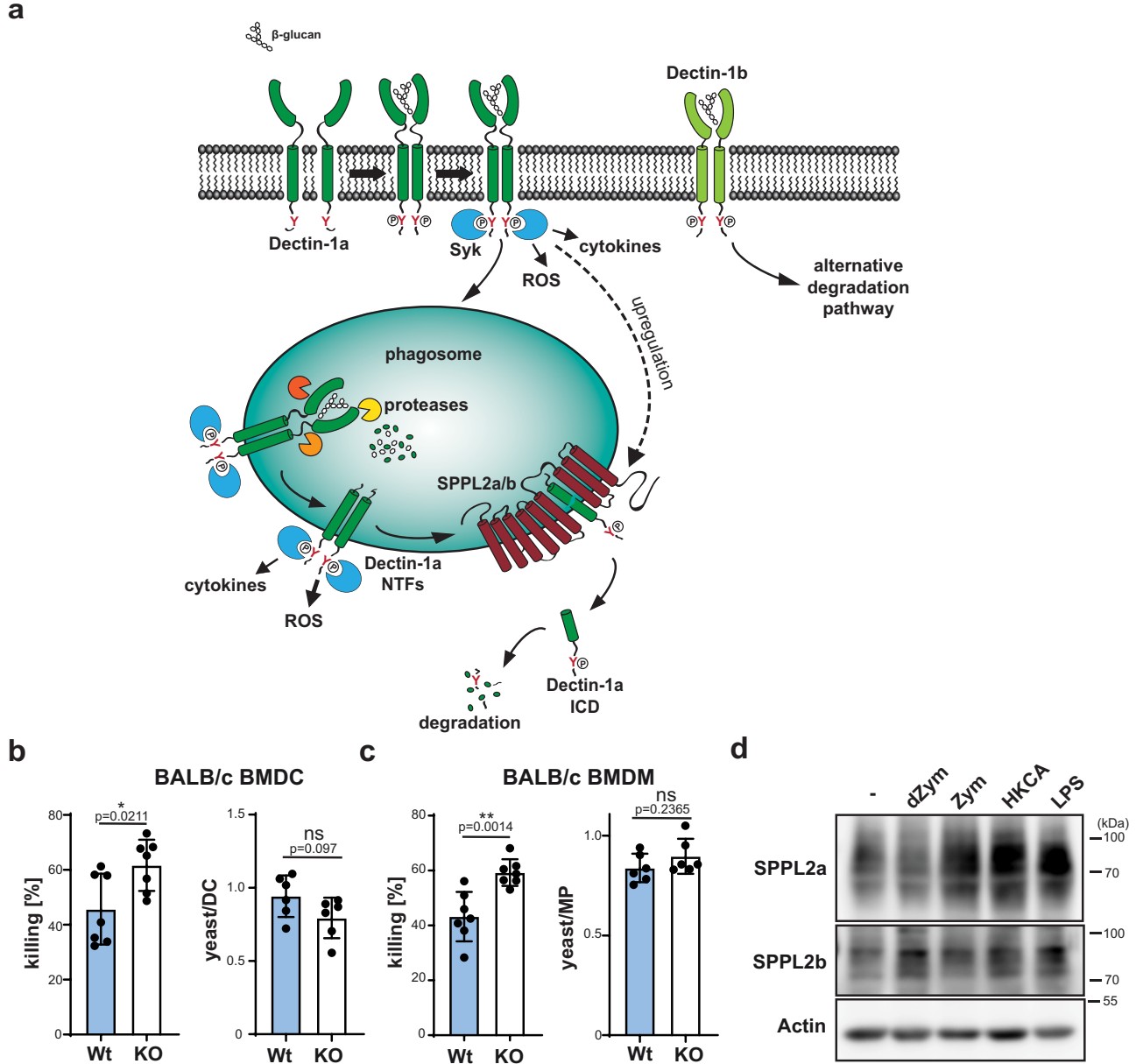

**Fig. 7 SPPL2 proteases regulate killing of *C. albicans* yeasts as part of a regulatory circuit. a** Schematic overview of differential processing of Dectin-1 isoforms. Upon ligand binding, Dectin-1a is phosphorylated at the intracellular hemITAM (red Y) leading to the recruitment of kinases including Syk as well as the induction of downstream responses including the formation of ROS and the induction of cytokine secretion. Additionally, ligand binding enforces internalisation of Dectin-1a into phagosomes where it is processed by soluble proteases to a membrane-embedded Dectin-1a NTF. This fragment is still capable of transmitting signalling responses and requires SPPL2a/b for its degradation and termination of Dectin-1a-dependent signal transduction. Dectin-1b utilises alternative degradative pathways. **b** Killing as well as phagocytosis of *C. albicans* yeasts (MOI 1) was assessed for BMDC from either wild type (Wt) or *SPPL2b*−/− mice (2b KO) mice on BALB/c background. Killing: $N = 3$, $n = 7$; phagocytosis $N = 2$, $n = 6$. Two-tailed unpaired Student's *t* test. **c** The same was done for BMDM derived from the same mice. Killing: $N = 3$, $n = 7$; phagocytosis $N = 2$, $n = 6$. Two-tailed unpaired Student's *t* test. **d** BMDCs from BALB/c wild-type mice were treated for 24 h with either 50 µg/ml dZym, 50 µg/ml Zym, MOI 10 HKCA or 500 ng/ml LPS and finally monitored for SPPL2a/b protein abundance by western blotting employing specific antibodies against the proteases. $N = 4$, $n = 4$. All diagrams depict Mean values ± SD.

fraction of full length Dectin-1a or the NTF also follows ESCRT-mediated degradation. However, a major amount of the NTF is segregated away from this sorting process and depends on intramembrane proteolysis for its turnover. Blocking this cleavage leads to NTF accumulation indicating that the MVB pathway cannot compensate. Cbl-b facilitated sorting and degradation of Dectin-1 was proposed to depend on the ubiquitination of cytosolic lysine residues[21–23]. The cytosolic domain does not differ between the two examined Dectin-1 isoforms and is

preserved in the NTF, which thus contains all proposed sorting determinants offering no explanation why it is spared from this process. Based on this, it seems possible that the NTF is actively retained in the membrane by a yet to be characterised process possibly involving the stalk domain.

The evolvement of the stable Dectin-1a NTF as active signalling component made an alternative mechanism for signalling termination necessary, which also provides an additional layer of regulation. Intramembrane proteases seem to be particularly well

suited for this purpose, since they are intrinsically slow enzymes[51]. Thus, clearance of the NTF may follow its generation with a certain latency, which is consistent with the idea of extending receptor signalling by this means. Cleavage of the Dectin-1a NTF by SPPL2a/b releases the intracellular domain into the cytosol. We failed to detect this cleavage fragment reliably under overexpression as well as endogenous conditions, which is in agreement with findings for several other SPPL2a/b substrates[28]. This suggests that the Dectin-1a intracellular domain is rapidly further degraded once liberated from the membrane, why we have not looked into potential functions of this fragment. Overexpression or knockout of SPPL2a/b did not modulate levels of full-length Dectin-1 reflecting that SPPL2a/b usually require substrates with a short ectodomain[28]. Therefore, the proteolytic processing of the Dectin-1a ectodomain in combination with the NTF retention in the phagosomal membrane turns this protein into a SPPL substrate.

SPPL2b is primarily localised to the plasma membrane[30,31]. However, as Fig. 3e shows SPPL2b is actively recruited to Dectin-1a containing phagosomes in addition to SPPL2a, which resides in endosomal/lysosomal compartments anyway[30]. Our data suggest that both SPPL2a and SPPL2b contribute to cleavage of the Dectin-1a NTF as we have demonstrated for the related CLR LOX-1[26]. In HEK293 cells the contribution of SPPL2a appeared to be rather small. However, in BMDCs and BMDMs this may be much bigger based on the additional NTF accumulation in $SPPL2b^{-/-}$ cells induced by InX. This could indicate that the contribution of both proteases varies in a cell type-dependent manner. However, we can also not exclude that the InX effect in SPPL2b-deficient cells is in part indirect. The CD74 NTF accumulating upon SPPL2a inhibition has been shown to alter endocytic trafficking and it is possible that this indirectly influences trafficking of the Dectin-1 NTF and/or recruitment of SPPL2b. Based on NTF levels, it may be assumed, that changes of Dectin-1 induced downstream responses like ROS and cytokine production may be even more pronounced in cells lacking both proteases SPPL2a and SPPL2b as compared to the SPPL2b single-deficient cells. Unfortunately, this cannot be easily assessed, since upon loss of SPPL2a accumulating CD74 NTFs interfere with trafficking and signalling in immune cells, in particular of Dectin-1 itself[43,44,52]. Therefore, cytokine[44] and ROS (Supplementary Fig. 7d) responses are disturbed in SPPL2a-deficient BMDCs. Though most of these changes are reversed in SPPL2a-CD74 double-deficient mice[25,43,44], this happens at the price of losing CD74, which based on its pleiotropic functions[53] in particular with regard to MHCII trafficking, has major consequences in immune cells. Therefore, although SPPL2a/SPPL2b/CD74 triple KO mice on a BALB/c background may theoretically be of interest, in light of the described interpretation difficulties the gain of such a model may be questioned. Though the phenotype of $SPPL2b^{-/-}$ BMDCs may underestimate the full impact of intramembrane proteolysis on Dectin-1 signalling, it provides an important proof-of-concept model and is associated with pathophysiologically relevant effects on phagocyte killing capacity.

SPPL2b deficiency particularly enhanced ROS production induced by artificial Dectin-1 ligands as well as viable *C. albicans*. In contrast, the augmentation of the analysed cytokine responses was more variable which correlated with their Dectin-1 dependence. This supports our model, since SPPL2b deficiency is only expected to enhance the Dectin-1 associated part of the overall signalling response. Though the efficiency of the antibody-blocking approach to assess Dectin-1 dependence may have limitations, the observed blocking of ROS responses by ~50% confirms its general applicability. Also in a previous report, ROS production to Zym seemed to be more strongly affected by loss of

Dectin-1 than induction of certain cytokines[9]. It may not be surprising that complex pathogens like *C. albicans* in native or heat-killed form are recognised by a plethora of PRRs from different families[54], which cooperatively shape the resulting cellular response. In addition to Dectin-1, the yeast cell wall preparation Zym also stimulates TLRs[39]. Alkali treatment was suggested to deplete the TLR-activating determinants while preserving the Dectin-1 binding β-glucans[39]. Gantner et al. mainly validated this based on NFκB activation in RAW cells[39]. Our results suggest that dZym can also trigger Dectin-1-independent cytokine responses in primary immune cells, however, ROS production was clearly Dectin-1-dependent to a major degree. Altogether, the contribution and requirement of Dectin-1 for the induction of cytokine responses to Zym[7] and *C. albicans*[7,9,55] varies between different cytokines and studies. A specific limitation in correlating our data with the literature is that commonly cells from BL6 mice and not like in our case from BALB/c mice have been analysed. It seems likely that differences in Dectin-1 dependence between our data and previous reports at least in part can be attributed to this difference in genetic background. In line with this, Dectin-1 deficiency had very diverging effects on outcome and cytokine profiles in an in vivo *C. albicans* infection model on BL6 and BALB/c backgrounds[56].

Our data are consistent with a specific modulation of Dectin-1 signalling by SPPL2 proteases as treatment with the Dectin-1 blocking antibody in most cases abolished the effect of protease-deficiency. However, as the majority of CLRs are type II trans-membrane proteins, it is tempting to speculate that also other members may be subjected to a similar regulation by intra-membrane proteolysis. In this line, we observed SPPL2-dependent turnover of NTFs derived from overexpressed Mincle, Dectin-2 and DC-SIGN, which can also contribute to anti-fungal responses[54,57]. Existence and proteolysis of the fragments at the endogenous level remains to be confirmed. Further work would also need to address to what extent in these cases ligand binding, receptor internalisation and NTF generation are coupled like for Dectin-1. As we observed previously for the oxidised LDL receptor LOX-1, also constitutive, ligand-independent processing can lead to the production of NTFs which then can either modulate signalling responses of the full-length receptor or have an NTF-intrinsic signalling capacity[26]. However, accumulating NTFs may not always exert a signal amplification. It is also possible that NTFs interfere with the function of the full length receptors they are derived from like in the case of TREM2[58] and diminish signalling. Such a model may also be considered when analysing further the reduced fungal responses we observed in SPPL2b-deficient BMDCs on a BL6 background, which are currently unexplained.

Importantly, activation of BMDCs by Dectin-1 and other PRRs led to an upregulation of SPPL2a and SPPL2b. Thus, PRRs switch on a regulatory circuit, which can negatively regulate the Dectin-1 signalling response, limit cellular activation and prevent over-shooting immune responses. Since upregulation of SPPL proteases occurs with a certain latency, this mechanism seems particularly suited to control PRR signalling and associated pathways when the immune system has been primed already and further activation does not provide an additional benefit. This could help to balance the killing capacity of immune cells with ROS-induced tissue damage over the course of an infection. Taken together, our findings complement our current view on Dectin-1-dependent signal transduction and its regulation. As indicated above, this is of special relevance since most existing studies are based on mouse models lacking expression of the Dectin-1a isoform, which is a major isoform in humans. Therefore, these findings close a relevant gap how anti-fungal immune responses are fine-tuned.

## Methods

**Mouse lines**. SPPL2b-deficient mice on a C57/BL6 N *Crl* background have been described before[31]. To study SPPL2b-deficiency in a Dectin-1a expressing mouse strain, this allele was backcrossed into a BALB/cAnNCrl background for 10 generations. Breeding and sacrificing of mice was approved by the Ministerium für Energiewende, Landwirtschaft, Umwelt und ländliche Räume of Schleswig-Holstein (V 242.7224.121-3) and the Landesdirektion Sachsen (TV A 12/2018, DD24.1-5131/450/12). Mice were kept in individually ventilated cages in the animal facilities of the Christian-Albrechts-University Kiel or within the Medizinisch-Theoretisches Zentrum, TU Dresden with a 12 h/12 h dark-light cycle at ambient temperatures of 22–24 °C and relative humidity of 50–55%.

**Plasmids**. Expression constructs of the murine and human Dectin-1 isoforms were generated in pcDNA3.1/Hygro+, pcDNA5/FRT (Thermo) or the retroviral vector pMSCV-puro using standard cloning techniques. Expression constructs of murine[31] and human[59] SPPL proteases have been described before. A GFP-tagged expression construct of the kinase Syk was kindly provided by Carlos del Fresno Sánchez, Centro Nacional de Investigaciones Cardiovasculares Carlos III, Madrid, Spain[42]. Murine N-terminally HA- and C-terminally FLAG-tagged Dectin-2 and Mincle were cloned from BMDC cDNA.

**Cell lines and transfection**. HeLa and HEK293 cells were purchased from DSMZ, Germany. J774.E cells were provided by Albert Haas, Bonn, Germany. Isolation of wild type as well SPPL2a/b double-deficient murine embryonic fibroblasts (MEFs) had been described in ref. [26]. Cells were cultivated in DMEM supplemented with 10% fetal calf serum (FCS) (Thermo Fisher) as well as 100 U/ml penicillin (Sigma) and 100 μg/ml streptomycin (Sigma) in a humidified 95% air/ 5% CO2 atmosphere.

T-Rex™-293 (HEK293) cells were purchased from Invitrogen and were cultivated in DMEM with Glutamax (Invitrogen) supplemented with 10% fetal calf serum (Sigma), 1% penicillin/streptomycin (Gibco) and 5 μg/ml Blasticidin (Invitrogen) in a humidified 95% air/ 5% CO2 atmosphere. Cells were incubated in antibiotic free medium containing 1 μg/ml doxycycline for induction of SPPL2 expression and harvested 24 h after transfection. SPPL2a knock out cells have been described earlier[60].

Stable cell lines expressing murine HA-Dectin-1a-FLAG and a correspondingly tagged 1b isoform were generated by transfecting Flip-In™ 293 cells (Invitrogen) according to manufacturer's instructions. After hygromycin selection, surviving cells were maintained and used for experiments without subcloning.

As indicated in the figure legends, cells were treated with the following compounds: E-64D (40 μM, Enzo), Leupeptin (2.5 μM, Roth), AEBSF (500 μM, Sigma), Bafilomycin A1 (100 nM, Sigma), (Z-LL)₂-ketone (40 μM, Peptanova), inhibitor X (1 μM, Tocris), Latrunculin A (5 μM, Enzo), Cytochalasin D (1 μM, Sigma) and DAPT (10 μM, Sigma). Except of AEBSF and Leupeptin, which were dissolved in sterile water, all other compounds were applied as DMSO-based solutions. To analyse processing and downstream functions of Dectin-1 as well as other CLRs and TLR4, the following ligands were applied: Zymosan (Zym, Sigma), depleted Zymosan (dZym, Invivogen), Curdlan (Invivogen), lipopolysaccharide (LPS) from *E. coli* (Sigma), heat-killed *Candida albicans* (HKCA, Invivogen) and heat-killed *Mycobacterium tuberculosis* (HKMT, Invivogen). Cells were transfected with Turbofect (Thermofisher), Lipofectamine 2000 (Thermofisher) or polyethylenimine (PEI).

**siRNA transfection of HEK cells**. HEK cells stably overexpressing HA-mDectin-1a-FLAG were transfected with specific siGenome smartpool siRNAs (Thermo Fisher) targeting either human SPPL2a (GUUGUUGCCUGGAGACGUA, GGA-GUGGACUAGUUGAAUU, CCUCAUGCCUGUUUCAAUA and GGUAACA GCUAUCAGAUGA) or SPPL2b (GUACAGUCCUCCAGGGUAU, GCUCUAC UAUUUCUACGAU, GACGCAGUAUGAUGAGAAU, GGAACUGCACCUUC UAUGA) at a final concentration of 20 nM. As control, the siGENOME Non-Targeting siRNA Pool #1 was employed (UAGCGACUAAACACAUCAA, UAAGGCUAUGAAGAGAUAC, AUGUAUUGGCCUGUAUUAG, AUGAACG UGAAUUGCUCAA). Transfection was performed using INTERFERin transfection reagent (Polyplus transfection) according to the manufacturer's recommendations. To ensure efficient downregulation of SPPL2a/b expression, the siRNA transfection was repeated 48 h after the initial transfection. Cells were harvested 96 h after the first transfection.

**Retroviral transduction**. J774.E or MEF cells were stably transduced with eco-tropic retroviruses. For this purpose, Platinum-E cells were transfected with pMSCV constructs. Virus-containing media were harvested after 48 h, filtered (20 μm, BD Biosciences) and transferred to the respective cells after supplementation with 8 μg/ml polybrene. To increase transduction efficiency, plates were centrifuged for 90 min at 1200 g at room temperature. Selection of transduced cells was performed with 10 μg/ml puromycin (Invivogen) starting 1 day after transduction.

**Generation of bone marrow-derived dendritic cells and macrophages (BMDMs)**. BMDCs were generated as described before[31] based on the procedure by Lutz et al.[61]. Red bone marrow was flushed from femur and tibia and dissociated

by passing through a 23G cannula and a 100 μm cell strainer. In all, $5 \times 10^6$ cells were seeded in 10 mL BMDC medium (RPMI 1640 medium supplemented with 10% heat-inactivated fetal bovine serum (Thermo), 100 U/ml penicillin (Sigma), 100 μg/ml streptomycin (Sigma) and 50 μM β-mercaptoethanol (GIBCO). Differentiation of cells was initiated by 20 ng/mL recombinant murine GM–CSF (Immunotools) directly after seeding (day 0). Fresh medium was supplied to the cultures at day 3 and 6. At day 8, cells were harvested so that suspension cells and adherent cells, which were detached using Accutase, were combined prior to seeding cells for experiments.

For generation of BMDMs, $1 \times 10^7$ bone marrow cells collected as described above were cultivated for a total of 7 days in BMDM medium (DMEM supplemented with 20% heat-inactivated fetal bovine serum, 100 U/ml penicillin, 100 μg/ml streptomycin and 50 ng/ml rm M-CSF (Immunotools). After 3 days of culture, 5 ml of M-CSF-supplemented medium were added to each culture plate. For harvesting, cells were washed with sterile PBS and subsequently detached using Accutase.

**Stimulation of immune cells with PRR ligands**. For analysis of ligand-induced Dectin-1 processing or receptor-dependent signalling, $2 \times 10^6$ cells were seeded in a volume of 1 ml in six-well plates and subsequently treated with Zymosan (Sigma), depleted Zymosan (Invivogen), Curdlan (Invivogen), LPS from *E. coli* (Sigma) or heat-killed (HKCA, Invivogen) or living *Candida albicans* in the concentrations indicated in the respective figure legends. Finally, cells were harvested for western blot analysis. In order to determine cytokine levels upon stimulation of BMDCs, $0.5 \times 10^6$ cells were transferred to 24 well plates in a total volume of 1 ml and treated as indicated for 24 h. Supernatants were harvested by repeated centrifugation at $210 \times g$ and then $18,000 \times g$ for 10 min. Levels of either murine or human TNF, IL-1β, IL-23 and IL-6 were analysed employing the respective DuoSet ELISA kits (R&D). For measurement of cytokine secretion of BMDM, $0.25 \times 10^6$ cells were seeded in a volume of 500 μl per well of a 24-well plate. When Dectin-1 specificity of the respective readouts was assessed, cells were pre-incubated with 10 μg/ml anti-Dectin-1 (clone 2A11, BioRad) for 60 min prior to administration of the respective ligands.

**siRNA transfection of BMDCs**. siRNA duplexes were transferred by electroporation as described earlier[62]. In brief, $2 \times 10^6$ BMDCs were transfected with 0.4 nmol of either siGENOME Non-Targeting siRNA Pool #1 (UAGCGACUAAAC ACAUCAA, UAAGGCUAUGAAGAGAUAC, AUGUAUUGGCCUGUAUUAG, AUGAACGUGAAUUGCUCAA) or siGENOME Mouse Sppl2b siRNA SMART-pool (GUAAUGUGCUGCUUCAUGC, GAAUUUACUUUGUGGCCUG, GUC AUGAUCUGCGUGUUUG, GCCUCCAAGUGAAGAGCUA) using a Gene Pulser Xcell™ (BioRad). After electroporation, cells were incubated for additional 75 h before further treatment and analysis. Knockdown efficiency was controlled by western blotting.

**PBMC isolation**. Leucocyte concentrates from healthy adult blood donors were obtained from the Transfusion Medicine of the University Hospital Schleswig-Holstein (UKSH) in Kiel. Written informed consent was obtained from all blood donors, and the research was approved by the relevant institutional ethic committee of the CAU Kiel (D517/15). Blood samples were diluted to a final volume of 35 ml with buffer I (phosphate-buffered saline solution (PBS) + 2 mM EDTA) and added to 15 ml of Ficoll (PAA) in a 50 ml Falcon. Subsequently, the sample was centrifuged for 40 min at 400 g at RT. The upper phase was removed and the PMBC containing layer was collected using a sterile Pasteur pipette. PBMCs from two Falcons from the same donor were pooled in a fresh 50 ml Falcon, volume was adjusted to 50 ml using buffer I and cells sedimented for 10 min at 300 g at RT. Pellets were resuspended in 5 ml buffer I, again two Falcons from the same donor were pooled and cells spun down for 10 min at $200 \times g$ at RT and used for RNA isolation.

Alternatively, PBMCs were isolated from blood samples obtained with written informed consent from healthy donors by the German Red Cross Blood Donation Service North-East, Dresden. The study was approved by the local institutional review board of the Faculty of Medicine Carl Gustav Carus, TU Dresden, Germany (EK138042014). Blood was diluted 1:1 with PBS, layered on Ficoll-Hypaque solution (Biochrom) and density centrifugation was performed for 20 min at $980 \times g$ at RT. The interphase was harvested and PBMC were washed twice with cold PBS, resuspended in RPMI medium (Pan-Biotech) supplemented with 10% FCS (Sigma-Aldrich), counted and adjusted to $5 \times 10^5$ cells/ml for the analysis of antifungal responses. For ELISA measurements, 200 μl of these suspensions were plated into 96-well plates. Cells were treated in triplicates with the inhibitors and stimuli depicted in the respective Figure legends.

**Flow cytometry**. For staining of HEK cells stably overexpressing either HA-mDectin-1a-FLAG or HA-mDectin-1b-FLAG, cells were detached using Accutase and washed once with FACS buffer (2% FCS, 2 mM EDTA in PBS). For staining, cells were resuspended in 100 μl FACS buffer containing anti-Dectin-1-PE (RH1, Biolegend) and stained for 1 h on ice. Afterwards, cells were recovered by centrifugation and taken up in 100 μl FACS buffer containing 250 ng/ml propidium iodide (Invitrogen) prior to a final washing step with FACS buffer. Finally, cells

were washed once, resuspended in 200 µl FACS buffer and analysed by flow cytometry using a FACS Canto II. Data processing was performed using FlowJo software (BD Biosciences). In the respective histograms, PE signals of PI negative cells are depicted. The applied gating scheme is depicted in Supplementary Fig. 10.

**Protein extraction and western blot analysis**. Cells were washed with PBS. Harvested cells were sedimented and resuspended in lysis buffer (50 mM Tris/HCl, pH 7.4, 150 mM NaCl, 1.0% Triton X-100, 0.1% SDS) supplemented with Complete protease inhibitor mix (Roche), 0.5 µg/ml Pepstatin A (Sigma), 4 mM EDTA and 4 mM Pefabloc® SC Protease Inhibitor (Roth). For detection of phosphorylated proteins, also PhosStop phosphatase inhibitor mix (Roche) as well as 20 mM β-glycerophosphate, 20 mM sodium fluoride and 4 mM sodium orthovanadate were added to the lysis buffer. Samples were further disrupted by sonication employing a Branson Sonifier 450 (Emerson Industrial Automation) and kept on ice for 1 h. Lysates were cleared by centrifugation and protein concentration was quantified with the Pierce™ BCA Protein Assay Kit (Thermo Fisher Scientific). Prior to SDS-PAGE, lysates were adjusted to final concentrations of 1% (w/v) SDS, 100 mM DTT, 125 mM Tris-HCL, pH 6.8, and 10% (v/v) glycerol and proteins denatured for 10 min at 56 °C. Separation by SDS-PAGE was performed according to Laemmli[63] followed by protein transfer to nitrocellulose[64]. For immunodetection, primary antibodies were diluted in 5% milk powder (Roth) dissolved in Tris-buffered saline containing 0.1% (v/v) Tween-20 (TBST) or I-Block (Thermofisher). Phospho-specific antibodies were diluted in 5% BSA (w/v) (Roth) in TBST. The monoclonal 3F10 antibody for detection of the HA epitope was obtained from Roche. Human Dectin-1 was detected employing a rabbit monoclonal antibody from Cell Signalling (clone E1X3Z) Generation of polyclonal antisera against the N-terminus of murine Dectin-1[44], murine SPPL2a[30], human SPPL2a[65], murine SPPL2b[31] and human SPPL2b[26] has been described previously. For detection of Dectin-1 induced signalling, monoclonal antibodies against phosphorylated ERK1/2 (T202/Y204, D13.14.4E) as well as total ERK1/2 (137F5) from Cell Signalling were employed. Anti-FLAG M2 was purchased from Sigma-Aldrich. Anti-Syk (D3Z1E) and anti-GFP (D5.1) were obtained from Cell Signalling. For detection of murine or human CD74 antibodies from BD Biosciences (clone In-1) or Stress-Marq Biosciences (PIN.1) were applied. To confirm equal protein loading of membranes, polyclonal anti-actin (Sigma), monoclonal anti-cofilin (D3F9, Cell Signalling), polyclonal anti-EEF2 (ab33523, Abcam), monoclonal anti-αTubulin (2144 S, Cell Signalling) or anti-Calnexin (Enzo Life Sciences) were used. Horseradish peroxidase-coupled secondary antibodies were obtained from Dianova and Promega. Chemiluminescent signals were recorded with a LAS4000 imaging system (GE Healthcare) and images were processed with Adobe Photoshop or GIMP. Where applicable, signals were quantified densitometrically using ImageJ. For analysis of MAP kinase activation, first levels of phosphorylated ERK1/2 were divided by those of total ERK1/2 of the same sample and normalised to the ratio calculated for the 20 min time point of the Dectin-1a expressing cell line.

When harvesting T-Rex™-293 (HEK293) cells for analysis, cells were sedimented and resuspended in hypotonic buffer (10 mM Tris, 1 mM EDTA, 1 mM EGTA, pH 7,4) supplemented with protease inhibitor mix (Sigma Aldrich). Membranes were recovered by centrifugation at $16,000 \times g$ and samples were resuspended in a mix (1:1) of basic buffer (40 mM Tris-HCl, pH 7.8, 40 mM potassium acetate, 1.6 mM magnesium acetate, 100 mM sucrose, 0.8 mM DTT) and sample buffer (20% (v/v) glycerol, 3% (w/v) SDS, 3% (w/v) DTT dissolved in 0.5 M Tris, 0.8% (w/v) SDS, pH 6.8). Proteins were denatured for 10 min at 65 °C and separated by a modified Tris-Tricine gel[66], followed by protein transfer to PVDF membranes. For detection of several proteins from the same membranes, especially for detection of total ERK1/2 following detection of phosphorylated ERK1/2, bound antibodies were stripped off by incubating the membranes in glycine stripping buffer (100 mM glycine, 20 mM magnesium acetate, 40 mM KCl, pH 2.2) for 45 min at 60 °C. Afterwards, membranes were washed repeatedly with TBST, blocked with 5% milk powder in TBST for 1 h and then incubated with the respective new antibody as described above.

**Immunoprecipitation**. Cells were harvested as described above, however, cell lysates were prepared in IP buffer (50 mM Tris-HCl, 150 mM NaCl, 1% (w/v) Triton X-100, 4 mM EDTA) supplemented with the same proteinase and phosphatase inhibitors as in the regular cell lysis buffer. For affinity purification, 1000–1500 µg of protein were adjusted to a final volume of 500 µl in IP buffer and supplemented with 2 µl anti-Syk (D3Z1E, Cell Signalling) or anti-phosphorylated Tyrosin (pY99, Santa Cruz biotechnologies) and incubated on a rotating wheel over-night at 4 °C. The subsequent pulldown was facilitated using 25 µl protein G agarose beads (Thermo) for 2 h under the same conditions. After five washes with IP buffer, proteins were eluted from the beads by boiling in SDS-PAGE sample buffer at 95 °C for 5 min. Analysis of precipitated proteins was carried out by western blot analysis as indicated above.

**RT-PCR**. RNA was isolated with Nucleospin RNA Kit (Macherey Nagel) following the supplier's protocol and reverse-transcribed with RevertAid Reverse Transcriptase (Thermo Scientific). For qualitative analysis of Dectin-1 isoform expression in murine and human cells, 1 µl of cDNA was employed for subsequent PCR analysis based on the following primers: mDectin-1 fw: ATGAAATATCACTCT

CATATAG; mDectin-1 rev: TTACAGTTCCTTCTCACAGATAC; hDectin-1 fw: GGAATATCATCCTGATTTAGAAAATTTGG; hDectin-1 rev: CATTGAAA ACTTCTTCTCACAAATACTATATG; hGAPDH fw: GGAAAGCTGTGG CGTTGGCGTGAT; hGAPDH rev: CTGTTGCTGTAGCCGTATTC. To validate knockdown of SPPL2a and SPPL2b expression in HEK cells, qPCR analysis was carried out employing the Universal Probe Library System from Roche using the CFX384 Touch™ Real-Time PCR Detection System (Biorad) based on the following primers: hSppl2a: CTTGCACACTTATTACTGCCTCA, CAATC CAAATGGTCCATCATC, probe #80; hSppl2b: TCCTGGGTTTCGGAGACAT, CAAACCTGTGGCAGTAGGC, probe #66; hTuba1c: CCCCTTCAAGTTCTAG TCATGC, GCATTGCCAATCTGGACAC, probe #58.

**Biotinylation of cell surface proteins**. PBMCs from one donor were divided into two aliquots and washed two times with PBS-CM (137 mM NaCl, 2.7 mM KCl, 10 mM Na₂HPO₄, 1.8 mM KH₂PO₄, 0.1 mM CaCl₂, 1 mM MgCl₂, pH 8.0). Afterwards, one aliquot was resuspended in PBS-CM (- Biotin), while the other was incubated in PBS-CM supplemented with 1 mg/ml membrane-impermeable (Sulfo-NHS-SS-Biotin, Thermo Scientific). Cells were incubated for 30 min on ice. After centrifugation at 210 g and 4 °C for 10 min, cells were resuspended in quenching solution (50 mM Tris in PBS-CM, pH 8.0) to eliminate unreacted biotin. After 10 min, PBMCs were washed twice with PBS-CM and then lysed as described above. A small portion of the lysate was saved as input sample. Equal protein amounts were subjected to pull-down of biotinylated proteins employing 50 µl High Capacity Streptavidin Agarose beads (Pierce) for 1 h at 4 °C. Following centrifugation of the bead suspension, supernatants were saved as unbound fraction. Beads were washed five times with 1 ml lysis buffer (as described above) and finally proteins were eluted from the beads by incubation with 1x SDS sample buffer at 56 °C for min. Samples were subsequently analysed by western blotting.

**Indirect immunofluorescence**. Cells adherent to glass coverslips were fixed with 4% (w/v) paraformaldehyde. The staining procedure has been described in[64]. Anti-HA (3F10, Roche), anti-Myc (9B11, Cell Signalling), a polyclonal rabbit antibody against an N-terminal epitope of murine Dectin-1[44] and anti-LAMP-2 (2D5)[67] were employed as primary antibodies which were then visualised by Alexa 488- and 594-conjugated secondary antibodies (Molecular Probes). DAPI (4-,6-diamidino-2-phenylindole from Sigma-Aldrich) was added to the embedding medium. Images were recorded with Olympus FV1000 or Leica SP5 confocal laser scanning microscope. Processing of images was performed with Olympus Fluoview, ImageJ and Adobe Photoshop software.

**Culture of Candida albicans**. C. albicans strain SC5314[68] was plated on YPD plates at 30 °C over-night. On the following day, a single colony was picked for over-night culture in 25 ml YPD medium at 30 °C shaking at 180 rpm. Next day, 25 ml of the respective culture were spun down for 5 min at 3000 g and washed twice with 50 ml sterile PBS. Finally, the washed pellet was resuspended in PBS, yeast cells counted using a Neubauer chamber and cell concentration adjusted to $10^9$ cells/ml. The resulting cell suspension was then used for stimulation of BMDCs in a multiplicity of infection (MOI) of either 0.5, 1, 5 or 10 depending on the respective experiment.

**Measurement of ROS production**. For measurement of ROS production, 30,000 BMDCs were seeded per well of a white 96 well plate (Greiner) in a volume of 50 µl in RPMI medium without FCS and phenol red (Gibco). Cells were allowed to settle for 1 h. To induce ROS production, cells were treated with 50 µg/ml depleted Zymosan, 50 µg/ml Zymosan, 200 µg/ml Curdlan, MOI 10 HKCA or 500 ng/ml LPS, which were added to the cell suspension in a total volume of 50 µl as a twofold concentrated solution. Additionally, these solutions were supplemented with 100 µM L-012 (Tocris) serving as luminescent probe for ROS detection. In blocking experiments, cells were incubated with 10 µg/ml anti-Dectin-1 (2A11, Biorad) for 30 min prior to ligand application. Luminescent signals were detected either employing Biotek Synergy, Infinite M Plex (Tecan), Victor³ 1420 (Perkin Elmer) or CLARIOstar Plus (BMG labtech) multi-mode readers every 2.5 min for a total of 90 min. ROS production was quantified as Area under the curve (AUC), which was calculated using GraphPad Prism software.

**Determination of fungal killing capacity and phagocytosis**. To determine phagocytosis of living C. albicans yeasts by BMDMs/BMDCs, $5 \times 10^5$ cells were allowed to adhere on cover slips for 2 h. Subsequently, cells were inoculated with living C. albicans in a MOI of 1. Afterwards, cells washed and fixed with 2% PFA for 15 min. Following two washing steps, non-internalised yeast cells were stained using 2 µg/ml Concanavalin A-AlexaFluor (Thermo Scientific) 647 at 37 °C for 30 min. Samples were washed twice with PBS and immune cells were lysed for 5 min by addition of 0.5% Triton X-100 in PBS. After three PBS washing steps, fungal cell walls were stained with 35 µg/ml Calcofluor White (Biotium) for 20 min at RT. Finally, samples were washed three times with ddH₂O and mounted on microscopy slides employing a mixture of Mowiol/DABCO as described above but without addition of DAPI. Phagocytosed yeasts per cell were counted based on pictures taken with a Zeiss Axioimager Z1 (Carl Zeiss).

Fungal killing was assessed by seeding $8 \times 10^4$ BMDMs or BMDCs into 96-well plates. Wells without immune cells served as control. After 1 h, cells were co-cultured with living *C. albicans* (MOI 1) for 2 h prior to lysis of immune cells by addition of Triton X-100 in a final concentration of 1% (w/v). Fungal cells were resuspended by vigorous pipetting. Subsequently, suspensions were diluted and plated on YPD agar plates which were then incubated for 2 days at 30 °C. Colony numbers obtained for each sample were counted. Survival rates were obtained by normalising the sample colony numbers to those calculated for the above-mentioned control wells. Killing rates were finally calculated by subtracting survival rates from 100%.

**Statistical analysis**. Data were analysed using Microsoft Excel. For statistical analysis, GraphPad Prism software was used. The statistical tests employed for the individual experiments as well as the respective numbers of experiments (*N*) as well as the number of independent samples/mice tested (*n*) are indicated in the Figure Legends. For analysis of ROS measurements, area under the curve (AUC) was calculated from the individual sample curves using GraphPad prism software. Threshold was set at the mean of the first y value. Data were normalised to the mean of the respective control/wild type samples and are displayed as fold changes. For PBMC experiments in which inhibitors were applied, values of InX- or DAPT-treated samples were directly normalised to that of the DMSO-treated sample from the same donor due to the high variation in response between the individual donors. To allow an assessment of absolute cytokine concentrations detected and the degree of cytokine induction observed upon treatment of the different employed immune cells with various ligands, data from representative experiments are displayed in Supplementary Figs. 11–14. All diagrams show mean values ± SD.

**Reporting summary**. Further information on research design is available in the Nature Research Reporting Summary linked to this article.

## Data availability

The authors declare that all data supporting the findings of this study are available within the article and its Supplementary Information files. Source data are provided with this paper.

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

## Acknowledgements

We thank Petra Peche, Sebastian Held and Doreen Ussath for excellent technical assistance and organisational support. We are grateful for Syk expression plasmids to Dr. David Sancho Madrid and Dr. Carlos del Fresno Sanchez, Centro Nacional de Investigaciones Cardiovasculares (CNIC), Madrid, Spain. Flag-hDectin-1a-V5 expression plasmid was kindly provided by Dr. Maryam Khojasteh-Fard and Prof. Christian Haass, Deutsches Zentrum für Neurodegenerative Erkrankungen (DZNE), Munich, Germany. We thank Prof. Albert Haas for providing J774 cells. Acquisition of microscopic images was performed within the Core Facility Cellular Imaging of TU Dresden. This work was supported by the Deutsche Forschungsgemeinschaft to B.S. (125440785/SFB877, project B7; 251390220 /SCHR 1284/1-1, SCHR1284/1-2; 380321491/SCHR1284/2-1) to R.F. (263531414/FOR 2290 and 254872893/FL 635/2-2) and to T.M. (431664610/ME 5459/1-1).

## Author contributions

T.M., A.Y.S.N., N.L., C.S., K.S., R.W., A.T., V.S., P.N., A.C.G. and J.W. performed the experiments and analysed data. V.S., P.N. and J.J. provided expertise in conducting siRNA-mediated knockdown in BMDCs. R.W., A.T. and M.S. provided human PBMCs for stimulation experiments. K.S., A.D. and I.D.J. helped to perform infection experiments with *C. albicans*. P.S. and S.R.J. provided general support. I.D.J., R.F., J.J. and M.S. gave conceptual advice. B.S. designed, conceptualised and supervised the research and analysed data. B.S. and T.M. wrote the manuscript. All authors contributed to the editing of the manuscript.

## Funding

## Competing interests

The authors declare no competing interests.
