## [Peer Review File · Nature Communications]

Phagosomal signalling of the C-type lectin receptor Dectin-1 is terminated by intramembrane proteolysisREVIEWER COMMENTS

Reviewer #1 (Remarks to the Author):

This study by Mentrup and colleagues entitled 'Phagosomal signaling of the C-type lectin receptor Dectin-1 is terminated by intramembrane proteolysis' reveals a potential novel Dectin-1a signalling pathway initiating from an NTF degradation intermediate of Dectin-1a residing in the phagosomal membrane, which prolongs Dectin-1a signaling. The authors identified SPPL2a and b as the proteases responsible for further degradation of this NTF fragment and thus acting as negative regulators of this pathway, which in turn may influence anti-fungal responses to *Candida albicans*. The study carefully divulges the importance of model system when studying Dectin-1, as Dectin-1a and 1b isoforms are expressed differently between mouse strains and the NTF pathway does not occur in Dectin-1b signaling. Moreover, Dectin-1a is the predominant Dectin-1 isoform in humans. The findings of the authors are novel and interesting to the innate immunity and fungal infection research fields.

However, a major revision will be required to validate and strengthen their claims. First, throughout the study the authors go back and forth between murine primary cells, murine cell lines and human cell lines that overexpress human or mouse proteins. This does not add to the coherence of the study and given the fact that the authors have antibodies available that detect endogenous murine Dectin-1 they could perform more work in relevant primary murine cells. Second, the claim that the NTF pathway increases Dectin-1a signalling is shown by increased ERK phosphorylation, but the effects on downstream effects of this event are not entirely convincing. In this respect, the authors should perform additional control experiments to support their claim.

Below are my suggestions on how this study could be improved.

Major comments

1. The authors show in Fig S2F that they have an antibody that recognizes endogenous Dectin-1b in BL/6 mice and they use this antibody in various experiments with Balb/c derived primary cells suggesting that it also recognises Dectin-1a. In this respect, it is strange that the authors perform multiple experiments in which they express epitope-tagged murine Dectin-1 in human HEK cells in order to look at what happens with the Dectin-1a and -1b isoforms. These experiments could be performed in primary murine cells using their antibody. In particular, Figs 1G-O, 1Q-R and the complete Fig S2 all are chemical inhibitor experiments in HEK cells that should be performed in primary murine cells. This way, murine Dectin-1 can be studied in its murine context and also in a cell type that is relevant to its innate immunity function.

2. Another peculiar choice the authors make is to study Dectin-1 in primary murine BMDCs while in other experiments using the J774 murine macrophage cell line. I understand that the authors need a Balb/c-derived cell line for their study (and J774 is Balb/c, would be good to mention that for the reader...) and perhaps no suitable Balb/c DC cell lines are available, but a simple solution could be to use primary BMDMs rather than BMDCs for their experiments. Being phagocytes, BMDMs are suitable for their study, and an additional advantage could be that the SPPL2 proteins might not affect CD74 in BMDMs. The authors should repeat at least a couple of their key BMDC experiments in BMDMs in order to align their data with the J774 observations.

3. The authors show in Fig4 that ERK is more phosphorylated in dZym-stimulated cells in which Sppl2 proteins are inhibited or absent as well as in Balb/c cells vs Bl/6 cells, in line with more NTF signaling in all of these settings. However, in Fig 5 the authors cannot consistently support these findings when looking at more downstream dZym-induced Dectin-1 signaling effects: ROS generation was affected accordingly upon dZym stimulation but cytokine secretion was not. To further substantiate their ERK and ROS observations the authors should perform additional control experiments in BL/6-derived cells in which these effects should not be present due to the lack of Dectin-1a. They should inhibit SPPL2 in BL/6 cells and look at dZym-induced ROS generation, and they should stimulate BL/6-derived Sppl2b^{-/-} cells with dZym to look at ERK phosphorylation and ROS generation. However, the cytokine data remain puzzling and suggest that cytokines may derive from signaling induced by full-length Dectin-1a instead of by the NTF. To investigate this the authors could express degradation-resistant Dectin-1a or the Dectin-1a NTF fragment in J774 macrophages to dissect differential signaling by these proteins. Alternatively, different thresholds of NTF may be needed to signal to either ROS generation or cytokine production, raising the possibility that the SPPL2a still present in Sppl2b^{-/-} cells is sufficient to block cytokine production but not ROS generation. The authors could test this hypothesis by treating Sppl2b^{-/-} cells with SPPL2 inhibitors.

4. In contrast to the cytokine observations upon dZym stimulation in Fig 5, the data in Fig 6 suggest that cytokines are regulated in an SPPL2b-dependent manner when infecting cells with *Candida albicans*. This is puzzling and the authors try to explain this by claiming in Fig 7 that in fact SPPL2 proteases affect not only Dectin-1 but also three other CLRs involved in *C. albicans* signaling. However, the data in Fig 7 is very premature and in my opinion should be removed. The western blots in Fig 7 are not convincing and an effect of SPPL2 proteases on these other CLRs would warrant an entire separate study rather than a couple of quick westerns in this manuscript. Instead, the authors should perform a similar control experiment as suggested above. They should perform the *C. albicans* infection in Bl/6-derived Sppl2b^{-/-} cells. Since these BL/6-derived cells do not have Dectin-1a this experiment will tell whether the observations in Balb/c-derived Sppl2b^{-/-} cells are SPPL2b effects on Dectin-1a or not.

5. The human data in Fig 8 should be supplemented with dZym stimulation as well as *C. albicans* infection experiments in PBMCs in which SPPL2 proteases were inhibited to support the claim that similar mechanisms act in human cells.

Minor comments

1. I personally believe that all WB densitometries can be removed from the manuscript. Western blotting is a semi-quantitative method of protein measurement, and importantly: nearly all claims that the authors want to support with the densitometries in fact are visible by naked eye on the blots themselves. Including the densitometries only raises questions on statistics (where do error bars and stats come from when you show limited number of bands?), and make the figures very crowded. I would suggest to remove them.

2. ELISA cytokine measurements should not be shown as normalized values. Please provide the absolute cytokines measured and include the unstimulated conditions in order to appreciate the levels of induction.

3. Somehow, Fig 1J is missing.

4. CLRs is a more accepted abbreviation than CTLRs for C-type lectin receptors.

Reviewer #2 (Remarks to the Author):

In this manuscript Mentrup et al. have investigated the mechanism by which Dectin-1 signaling is terminated. The investigators have noted that one of the two isoforms of Dectin-1 (Dectin-1a) seems to be cleaved upon activation of signaling. This cleavage seems to require internalization of the receptor into a phagosome, and the cleaved receptor appears to continue signaling until it is degraded. Degradation requires SPPL2 proteins that are capable of cleaving proteins embedded in membranes. Degradation by SPPL proteins, similar to what has been observed for Cbl-mediated degradation of the uncleaved intact proteins, is important to ultimately attenuate and terminate the signal. Overall, the study is clear and convincing.

Major comments/questions:

Supplemental Figure 1A (transfected HeLa cells) is not sufficient data to make the blanket statement that human Dectin-1b does not get to the cell surface. A single reference to another group, not included until the discussion, that also focuses on ectopically expressed receptor constructs, does not fill in the gaps. Surface labeling of primary cells would be more convincing. Or some other more quantitative assay aimed at expression of the native (not overexpressed) receptors. If this point is not important to the authors, a less definitive set of statements can be made.

The idea in the Dectin-1a/b signaling comparison in figure 2H-K is that somehow Dectin-1a signaling is faster (and thus “stronger”). Do the authors have any idea how the presence/role of an additional degradation pathway for Dectin-1a would make signaling appear faster (i.e. in the <10 min. range)? I could see how signaling might be sustained longer if the other degradation pathway (which will be SPPL2) might be slower and thus allow a longer duration of signaling, but this doesn't seem to be evident in this HEK experiment.

Is there evidence that the prolonged signal of the protease trimmed dimer is important? The implication is that the persistence of signaling by the cleaved receptor (that is ultimately going to be degraded and inactivated anyway) is some sort of evolved mechanism to regulate signaling via Dectin-1a differently than from Dectin-1b. The data suggest that in vitro NTF formation and SPPL2-mediated degradation isn't particularly important for cytokine production. It presumably doesn't regulate or affect phagocytosis since the process doesn't begin until after internalization. It may modestly enhance or prolong ROS production, but is there any evidence that this amount of change in ROS production would be meaningful? The alternative explanation is that this is just a series of events leading to the receptor's degradation, not a particularly relevant way by which it is regulated.

Minor points:

The manuscript is full of novel observations. But throughout, the authors should take care to not appear to claim to first observe things others have previously reported (e.g. colocalization of Dectin-1 with LAMP, colocalization/association with Syk, etc.). I am fully in favor of reporting data that repeat what others have seen, but these data are best reported as being “consistent with previous observations”, including appropriate references.

Supplemental 4B is noted to show that “upon inhibition with ZLL no NTF accumulation was observed”, although if the gel bands were quantified it looks like there would be some accumulation.

P. 24: “The lacking effect of siRNA against SPPL2a is in line with its failure to inhibit NTF

proteolysis presumably due to knockdown efficiency at the protein level (Fig. 3I,K, Suppl. Fig. 4G)." None of these figures address whether SPPL2a protein was knocked down well or not.

Is there a reason that figures 1-4 skip "J"?

Point-by-point reply:

Reviewer 1:

Reviewer #1 (Remarks to the Author):

This study by Mentrup and colleagues entitled ‘Phagosomal signaling of the C-type lectin receptor Dectin-1 is terminated by intramembrane proteolysis’ reveals a potential novel Dectin-1a signalling pathway initiating from an NTF degradation intermediate of Dectin-1a residing in the phagosomal membrane, which prolongs Dectin-1a signaling. The authors identified SPPL2a and b as the proteases responsible for further degradation of this NTF fragment and thus acting as negative regulators of this pathway, which in turn may influence anti-fungal responses to *Candida albicans*. The study carefully divulges the importance of model system when studying Dectin-1, as Dectin-1a and 1b isoforms are expressed differently between mouse strains and the NTF pathway does not occur in Dectin-1b signaling. Moreover, Dectin-1a is the predominant Dectin-1 isoform in humans. The findings of the authors are novel and interesting to the innate immunity and fungal infection research fields.

However, a major revision will be required to validate and strengthen their claims. First, throughout the study the authors go back and forth between murine primary cells, murine cell lines and human cell lines that overexpress human or mouse proteins. This does not add to the coherence of the study and given the fact that the authors have antibodies available that detect endogenous murine Dectin-1 they could perform more work in relevant primary murine cells. Second, the claim that the NTF pathway increases Dectin-1a signalling is shown by increased ERK phosphorylation, but the effects on downstream effects of this event are not entirely convincing. In this respect, the authors should perform additional control experiments to support their claim.

Below are my suggestions on how this study could be improved.

We appreciate the positive general judgement of this reviewer. As outlined below we have carefully tried to address the valuable suggestions including the requested control experiments.

Major comments

1. The authors show in Fig S2F that they have an antibody that recognizes endogenous Dectin-1b in BL/6 mice and they use this antibody in various experiments with Balb/c derived primary cells suggesting that it also recognises Dectin-1a. In this respect, it is strange that the authors perform multiple experiments in which they express epitope-tagged murine Dectin-1 in human HEK cells in order to look at what happens with the Dectin-1a and -1b isoforms. These experiments could be performed in primary murine cells using their antibody. In particular, Figs 1G-O, 1Q-R and the complete Fig S2 all are chemical inhibitor experiments in HEK cells that should be performed in primary murine cells. This way, murine Dectin-1 can

be studied in its murine context and also in a cell type that is relevant to its innate immunity function.

We have taken up this suggestion and repeated all experiments studying Dectin-1 processing with inhibitors at the endogenous level in murine BMDCs. The respective data are depicted in Fig. 1j,m,o,r and Suppl. Fig. 2d,f,h. Importantly, these new data support the previous conclusions from the overexpression experiments.

2. Another peculiar choice the authors make is to study Dectin-1 in primary murine BMDCs while in other experiments using the J774 murine macrophage cell line. I understand that the authors need a Balb/c-derived cell line for their study (and J774 is Balb/c, would be good to mention that for the reader...) and perhaps no suitable Balb/c DC cell lines are available, but a simple solution could be to use primary BMDMs rather than BMDCs for their experiments. Being phagocytes, BMDMs are suitable for their study, and an additional advantage could be that the SPPL2 proteins might not affect CD74 in BMDMs. The authors should repeat at least a couple of their key BMDC experiments in BMDMs in order to align their data with the J774 observations.

As suggested, we repeated all experiments as far as possible in murine BMDMs. With regard to the determination of cytokine response not all of the ligands we had employed previously to activate BMDCs were able to induce cytokine secretion in BMDMs. Furthermore, with those ligands, which had an activating effect, we mainly elicited TNF responses. Employing our system, secretion of IL-6 or IL-1 β was in the range of the detection limit upon application of the utilized stimuli, why these data are not depicted. Altogether, the results obtained in BMDMs lead to similar conclusions as the BMDC data with only very subtle differences indicating that the proposed mechanism is not limited to BMDCs and is relevant also in other cell types. The BMDM data can be found in Fig. 7, Suppl. Fig. 3; Suppl. Fig. 4, Suppl. Fig. 6, Suppl. 7 and Suppl. Fig.8. However, also BMDMs express CD74. As depicted in Suppl. Fig. 6b, even a 1 h treatment of BMDMs with inhibitor X causes a major accumulation of the CD74 NTF. Therefore, we expect these cells to be affected by inhibitory effects of this fragment similar to BMDCs.

3. The authors show in Fig4 that ERK is more phosphorylated in dZym-stimulated cells in which Sppl2 proteins are inhibited or absent as well as in Balb/c cells vs Bl/6 cells, in line with more NTF signaling in all of these settings. However, in Fig 5 the authors cannot consistently support these findings when looking at more downstream dZym-induced Dectin-1 signaling effects: ROS generation was affected accordingly upon dZym stimulation but cytokine secretion was not. To further substantiate their ERK and ROS observations the authors should perform additional control experiments in BL/6-derived cells in which these effects should not be present due to the lack of Dectin-1a. They should inhibit SPPL2 in BL/6 cells and look at dZym-induced ROS generation, and they should stimulate BL/6-derived Sppl2b^{-/-} cells with dZym to look at ERK phosphorylation and ROS generation. However, the cytokine data remain puzzling and suggest that cytokines may derive from signaling induced by full-length Dectin-1a instead of by the NTF. To investigate this the authors could express degradation-resistant Dectin-1a or the Dectin-1a NTF fragment in J774 macrophages to

dissect differential signaling by these proteins. Alternatively, different thresholds of NTF may be needed to signal to either ROS generation or cytokine production, raising the possibility that the SPPL2a still present in Sppl2b^{-/-} cells is sufficient to block cytokine production but not ROS generation. The authors could test this hypothesis by treating Sppl2b^{-/-} cells with SPPL2 inhibitors.

We agree that the differential impact of SPPL2b ablation on ROS and cytokine production is puzzling on first sight. This discrepancy was mainly observed with artificial Dectin-1 ligands like dZym, Zym and HKCA. However, with regard to the *C. albicans* co-culture experiments, results seem to be more aligned as SPPL2b enhances production of both ROS and several cytokines. We considered concepts that the signaling triggered by the full length receptor and that initiated by the NTF have a different quality and are not identical with regard to the downstream pathways being activated. By this means it could be possible that the impact of the NTF signaling on ROS production is greater than that of the full length receptor. However, there is no obvious way to separate these entities as NTF signaling without activation of the full length receptor cannot be triggered.

In the end, it seems that the answer to resolve the described discrepancy is much more trivial. An apparent uncertainty in the stimulation experiments which we performed to assess Dectin-1 signaling is the specificity of the ligands. We propose that SPPL2b deficiency enhances Dectin-1 signaling. However, to observe this effect in any downstream response this specific parameter needs to be Dectin-1-dependent – at least to a relevant degree.

We tried to assess to which extent either ROS production or cytokine secretion provoked by stimulation of BMDCs or BMDMs with the different Dectin-1 ligands is Dectin-1-dependent in our system. Apparently, the cleanest and most unambiguous system to do this would have been to compare cells from wild type and Dectin-1-deficient mice on a Balb/c background. Dectin-1 knockout mice on this background have been described in the literature. However, despite contacting several groups in Germany and Europe including Gordon Brown we were not able to find someone who still maintains this mouse line on this background. Therefore, we decided to extend the approach which was already part of the first manuscript version on a small scale. Thus, we performed the respective assays in presence of the established and commercially available Dectin-1 blocking antibody 2A11. Data from these experiments are depicted in Fig. 5i, Fig.6l-o, Suppl. Fig. 7g-j and Suppl. Fig. 8e,f. The efficiency of this approach is difficult to control. However, as we saw up to 50% reductions in ROS production with this antibody we assume a general functionality.

The outcome of these blocking experiments was that the downstream responses in many cases were much less Dectin-1-dependent than we had anticipated. Whereas ROS production was clearly Dectin-1-dependent, this varied much more for the cytokine responses. Altogether, those readouts which we found to be affected by SPPL2b deficiency were Dectin-1 dependent to a much higher degree than others.

In general, it is well accepted that stimulants like zymosan, HKCA and also viable *C. albicans* in addition to activating Dectin-1 signaling also bind to and activate other PRRs. In contrast, depleted zymosan was suggested to represent a fairly Dectin-1 specific ligand. However, our results indicate that this may depend on the respective response analysed and also the cell type. In BMDCs, cytokine secretion upon dZym application was not affected by the Dectin-1 blocking antibody strongly suggesting that at least the dZym preparations we had obtained from Invivogen are able to activate BMDCs by additional receptors. dZym is based on zymosan which is a biological

material and which is then chemically processed to deplete the TLR binding moieties. When comparing our results with the literature, a major obstacle is that Dectin-1 has been mainly studied in the context of BL6 mice. This could explain some discrepancies. However, also in the published BL6 data Dectin-1 dependence of different responses in response to different stimuli varies and is not entirely consistent between different studies. This probably reflects the complexity of the system.

In summary, we think that the results from these Dectin-1 blocking experiments provide an explanation why certain readouts such as cytokine secretion following treatment with dZym and HKCA are not altered in the SPPL2b-deficient cells. If a readout does not or only to a very minor degree depend on activation of Dectin-1, it cannot be expected to be modulated by loss of SPPL2b as this would according to our model selectively enhance Dectin-1 signaling but not that from other PRRs.

We also looked into the hypothesis that additional inhibition of SPPL2a may increase observed effects of SPPL2b deficiency on ROS production and cytokines. We tested that systematically for all the readouts we have analysed (Reviewer Fig. 1). Altogether, inhibitor X treatment did not change the picture and further increase the responses. This may on first sight be surprising as we see that this treatment can further enhance accumulation of the Dectin-1a NTF. However, SPPL2a inhibition also leads to an accumulation of CD74 fragments of which we know that it disturbs endosomal trafficking, delivery of Dectin-1 to the cell surface and also Dectin-1 dependent cytokine responses (Gradtke et al., J Immunol 2021). In addition, we now also analysed ROS responses in SPPL2a-deficient BMDCs (Suppl. Fig. 6d) which revealed that also Dectin-1 triggered ROS production is significantly compromised in these cells. Furthermore, we saw, that even very short treatments with inhibitor X already led to a significant accumulation of CD74 fragments. Therefore, we believe that these inhibitor experiments in the end are not informative as discrimination of effects due to further inhibition of Dectin-1 proteolysis and accumulation of CD74 NTFs, which counter-act Dectin-1 triggered responses, is not possible. Based on these doubts, we felt that adding all these data to the manuscript may be of limited use as they would consume a lot of space, in particular for explaining their limited meaningfulness. Therefore, we currently decided to limit presentation of functional inhibitor experiments to those with PBMCs, which were requested and where no alternative system is available. The PBMC data are shown in Suppl. Fig.5 and are aligned with Suppl. Fig. 6 which provides a general explanation why the use of inhibitors to analyse downstream responses is of limited value in PBMCs but also murine cells. Therefore, the extensive data sets from treating wild type and SPPL2b-deficient BMDCs and BMDMs with inhibitor X, which are summarised in this reply letter as Reviewer Fig. 1, are currently not part of the revised manuscript. However, as the data is available, we are ready to include this if this is preferred by reviewers and editors.

Reviewer Figure 1: Effects of application of Inhibitor X on ROS formation and cytokine secretion in Balb/c BMDM and BMDC. a Balb/c BMDC from wild type (Wt) or SPPL2b-deficient (2b KO) mice were pre-incubated with DMSO as control or 1 μ M InX for 30 min. Afterwards, ROS formation induced by application of the indicated stimuli was measured employing the luminescent L-012 probe. ROS were evaluated as area under the curve. Here as well as throughout the Figure bars indicate ratios of values of InX-treated samples to the respective DMSO control for each individual animal. The red line marks a InX/DMSO ratio of 1 which corresponds to no alteration by InX upon ligand stimulation. Balb/c BMDC from Wt or 2b KO mice were stimulated with the indicated ligands and secretion of TNF (b), IL-1 β (c) or IL-6 (d) were quantified by ELISA. e The experiment described in a) was repeated with Balb/c BMDM instead of BMDC. f Balb/c BMDM were treated with viable *C. albicans* and secretion of TNF was monitored by ELISA.

4. In contrast to the cytokine observations upon dZym stimulation in Fig 5, the data in Fig 6 suggest that cytokines are regulated in an SPPL2b-dependent manner when infecting cells with *Candida albicans*. This is puzzling and the authors try to explain this by claiming in Fig 7 that in fact SPPL2 proteases affect not only Dectin-1 but also three other CLRs involved in *C. albicans* signaling. However, the data in Fig 7 is very premature and in my opinion should be removed. The western blots in Fig 7 are not convincing and an effect of SPPL2 proteases on these other CLRs would warrant an entire separate study rather than a couple of quick westerns in this manuscript. Instead, the authors should perform a similar control experiment as suggested above. They should perform the *C. albicans* infection in BL/6-derived *Sppl2b*^{-/-} cells. Since these BL/6-derived cells do not have Dectin-1a this experiment will tell whether the observations in Balb/c-derived *Sppl2b*^{-/-} cells are SPPL2b effects on Dectin-1a or not.

As described under point 3, the performed blocking experiments suggest that cytokine responses to the commercially obtained dZym preparations in Balb/c BMDCs in our experimental system were mostly Dectin-1-independent. This finding was also a bit surprising to us. However, it provides an explanation why SPPL2b deficiency did not influence these responses. Similarly, in the case of cytokine responses to *C. albicans* (with the fungal strain and mouse background employed) the overall amplitude of the responses was not affected by the Dectin-1 blocking antibody. However, the antibody abolished or at least significantly diminished the difference between SPPL2b-deficient and wild type cells. We think that this confirms that at least the increment is caused by loss of SPPL2b is dependent on Dectin-1.

We followed the recommendation to analyse ROS and cytokine responses in wild type and SPPL2b-deficient BMDCs and BMDMs from mice with a BL6 background. This data can be found in Fig. 5, Fig. 6, Suppl. Fig. 7 and Suppl. Fig. 8. We found that none of the analysed responses was increased by absence of SPPL2b in BL6-derived cells. This confirms our model which predicts that SPPL2b deficiency selectively enhances responses to Dectin-1a which is hardly present in BL6 mice. An unexpected finding was that SPPL2b deficiency on a BL6 background rather reduced some of the analysed cellular responses to PRR stimulation. As Dectin-1 proteolysis is not altered in these cells (Fig. 3j, Suppl. Fig. 3k), this cannot contribute to this phenotype. However, we currently have no hypothesis how this may be linked to the proteolytic activity of SPPL2b. It may be that other CLRs are involved. But in some cases also LPS-triggered responses were affected which may indicate that even some more general pathways are affected - presumably by some currently unknown substrate of SPPL2b. This will definitely need to be addressed in future studies.

Regarding the analysis of SPPL2-mediated cleavage of other CLRs we totally agree that the functional consequences and physiological relevance of these findings are beyond the present story. However, with regard to the analysis of proteolysis we would not consider the presented data as preliminary. Any deeper analysis beyond the depicted overexpression experiments would require the generation of antibodies against the N-termini of these receptors, which are not available commercially and which would be needed to detect these N-terminal fragments at the endogenous level. This is quite a substantial commitment. Our feeling is that the observation that other CLRs may also be substrates of SPPL2b may provide a slightly broader perspective to the manuscript and its main findings. Furthermore, it may be a starting point to consider when investigating mechanisms why some of the analysed responses were lower in SPPL2b-deficient cells

on a BL6 background as compared to the wild type. Based on these thoughts, we were unsure if taking this data out completely is the ideal solution. We would suggest to move the data from the main figure to the Supplement (Suppl. Fig. 9) and we also have significantly shortened the corresponding description and discussion in the text. However, if in the eyes of the editors and reviewers the manuscript would benefit from taking these data out completely, this can easily be done.

5. The human data in Fig 8 should be supplemented with dZym stimulation as well as *C albicans* infection experiments in PBMCs in which SPPL2 proteases were inhibited to support the claim that similar mechanisms act in human cells.

We have performed experiments in PBMCs treated with an inhibitor against SPPL proteases. These cells were then stimulated with different Dectin-1 ligands as well as viable *C. albicans*. This data is shown in Suppl. Fig.5. The intrinsic problem with this approach is that no potent, SPPL2b-specific inhibitor is available. In our hands, the most efficient compound to inhibit proteolysis of Dectin-1 is inhibitor X, which however was initially characterized as a γ -secretase inhibitor and cross-inhibits most SPPL proteases. In this case, we tried to control potential effects going to back to inhibition of γ -secretase instead of SPPL proteases by performing all experiments in parallel with DAPT. DAPT also efficiently inhibits γ -secretase, but does not target SPPL proteases. Thus, comparing effects by these two inhibitors would make it possible to evaluate whether certain effects are caused by inhibition of SPPL proteases or of γ -secretase. However, the fact that inhibitor X targets other SPPL proteases in addition to SPPL2b makes this experiment in our eyes very difficult to interpret as explained above already. We have shown previously (Schneppenheim et al., *J Exp Med* 2013, Hüttl et al. *J Immunol* 2015, Gradtke et al., *J Immunol* 2021) that SPPL2a has a critical impact on the function of certain immune cells. This is caused by the role of SPPL2a in degrading N-terminal fragments (NTFs) of CD74, the invariant chain of the MHCII complex. We have characterized this so far primarily in B cells and dendritic cells. However, also macrophages express CD74 and accumulate CD74 NTFs upon SPPL2a inhibition as we show in Suppl. Fig. 6b. As analysed in Gradtke et al., *J Immunol* 2020 the accumulating CD74 NTF has a major negative impact on Dectin-1 responses. The most obvious effect is that it comes to mistrafficking of Dectin-1 leading to reduced Dectin-1 surface levels. This negatively regulates cytokine responses and, as we show here in this manuscript in Suppl. Fig. 6d, also ROS production following Dectin-1 activation. Using an unselective inhibitor that targets both SPPL2a and SPPL2b also these negative regulations of Dectin-1 signaling will be initiated. We confirmed that even a one hour treatment with inhibitor X leads to a well detectable accumulation of CD74 NTF in human PBMCs similar to murine BMDCs and macrophages. Therefore, this inhibitor is certainly not a suitable system to analyse the effects of SPPL2b inhibition as these will be confounded by a concurrent SPPL2a inhibition.

In light of these thoughts, we feel that the obtained data, which do not readily prove that our hypothesis is also valid in primary human immune cells should be interpreted with caution. In absence of an efficient and highly specific system to modulate SPPL2b activity in these cells while sparing SPPL2a, the respective question cannot be tested appropriately and validated in one or the other direction with the currently available tools. Therefore, the immediate relevance of our findings for human anti-fungal responses has to remain an open question for the moment.

Minor comments

1. I personally believe that all WB densitometries can be removed from the manuscript. Western blotting is a semi-quantitative method of protein measurement, and importantly: nearly all claims that the authors want to support with the densitometries in fact are visible by naked eye on the blots themselves. Including the densitometries only raises questions on statistics (where do error bars and stats come from when you show limited number of bands?), and make the figures very crowded. I would suggest to remove them.

We have followed this recommendation and removed the densitometric quantification from the manuscript.

2. ELISA cytokine measurements should not be shown as normalized values. Please provide the absolute cytokines measured and include the unstimulated conditions in order to appreciate the levels of induction.

The cytokine data which we depict are summarized from several independent experiments (indicated by a capital "N" number) of which each was performed with cells from n=3 or 4 mice per genotype, i.e. biological replicates. Our experience is that absolute levels of cytokines can vary slightly between different experiments, which may just reflect the complexity of the system or different charges of the commercially obtained stimuli. These differences are not big. However, when pooling datasets from different independent experiments in terms of absolute concentrations our impression was that this inherent technical variation in the absolute levels may in the end obscure clear relative differences between genotypes and/or treatments which are consistently seen throughout these experiments. Obviously, the increase we get by SPPL2b deficiency is not huge so that it can be masked by these slight variations in absolute stimulation levels which apparently occur despite vigorous standardization. Based on this problem, we have chosen the presentation of summary data after normalization as it was able to depict biological effects which were seen throughout the runs and it was still a way to show a compiled version of really a substantial number of independent experiments and biological experiments. Therefore, we felt that these normalized data summaries are a more valid and meaningful representation of the data we have generated as compared to depicting single experiments which would have been an alternative.

However, we totally agree that the normalised data presentation also leads to a loss of information as the absolute cytokine concentration and in particular the comparison between levels released by stimulated and non-stimulated cells is needed to judge the overall strength of the cellular activation. To serve all these different needs and to have the advantages of both presentations, we have kept the normalized data summaries in the main figures and corresponding supplementaries and have added a compilation of representative data-sets (Suppl. Fig. 10, Suppl. Fig. 11, Suppl. Fig. 12 and Suppl. Fig. 13) for each set-up which are derived from single experiments and depict absolute cytokine concentrations. By this means, we hope that all necessary information is contained within the manuscript in order to fully evaluate the presented data.

3. Somehow, Fig 1J is missing.

This has been corrected.

4. CLRs is a more accepted abbreviation than CTLRs for C-type lectin receptors.

This has been changed.

Reviewer #2:

In this manuscript Mentrup et al. have investigated the mechanism by which Dectin-1 signaling is terminated. The investigators have noted that one of the two isoforms of Dectin-1 (Dectin-1a) seems to be cleaved upon activation of signaling. This cleavage seems to require internalization of the receptor into a phagosome, and the cleaved receptor appears to continue signaling until it is degraded. Degradation requires SPPL2 proteins that are capable of cleaving proteins embedded in membranes. Degradation by SPPL proteins, similar to what has been observed for Cbl-mediated degradation of the uncleaved intact proteins, is important to ultimately attenuate and terminate the signal. Overall, the study is clear and convincing.

We thank the reviewer for the positive overall judgement. We specified in the following how we have implemented the helpful suggestions.

Major comments/questions:

Supplemental Figure 1A (transfected HeLa cells) is not sufficient data to make the blanket statement that human Dectin-1b does not get to the cell surface. A single reference to another group, not included until the discussion, that also focuses on ectopically expressed receptor constructs, does not fill in the gaps. Surface labeling of primary cells would be more convincing. Or some other more quantitative assay aimed at expression of the native (not overexpressed) receptors. If this point is not important to the authors, a less definitive set of statements can be made.

We have analysed the distribution and functionality of the human Dectin-1a and -b isoforms in PBMCs at the endogenous level. We have used two approaches: First, in a more indirect approach we have analysed to which extent the different isoforms are degraded upon treatment of the cells with zymosan since degradation is ligand-dependent and therefore only possible when receptor and ligand meet at the cell surface. Whereas Dectin-1a is degraded nearly quantitatively, only part of Dectin-1b is lost from the cells following ligand application. This could suggest that the majority of the cellular pool of Dectin-1a is at the cell surface, whereas only a fraction of the Dectin-1b pool is accessible to zymosan and may therefore reside in intracellular compartments. This picture was confirmed by biotinylation and isolation of cell surface proteins using a membrane-impermeable biotinylation reagent. Dectin-1a was quantitatively recovered in the cell surface fraction. In contrast, a certain amount of Dectin-1b seems to reside intracellularly as it was not recovered by the streptavidin pulldown. However, the endogenous situation is indeed different from the overexpression experiments, where human Dectin-1b is entirely retained in the ER. A significant part of endogenous human Dectin-1b does reach the cell surface. These new experimental data are now depicted in Suppl. Fig. 1b,c. As we also state in the text, this indicates that in humans both isoforms do play a functional role. Thus, the role of Dectin-1a in humans is indeed not as predominant as we initially thought and it was certainly a great suggestion to have that clarified. Nevertheless, Dectin-1a constitutes about half of the total Dectin-1 pool at the plasma membrane of human PBMCs. Therefore, also in light of the new findings, BL6 mice, where this isoform is hardly present, do not represent a suitable model for the human situation.

The idea in the Dectin-1a/b signaling comparison in figure 2H-K is that somehow Dectin-1a signaling is faster (and thus “stronger”). Do the authors have any idea how the presence/role of an additional degradation pathway for Dectin-1a would make signaling appear faster (i.e. in the <10 min. range)? I could see how signaling might be sustained longer if the other degradation pathway (which will be SPPL2) might be slower and thus allow a longer duration of signaling, but this doesn’t seem to be evident in this HEK experiment.

In our initial analysis of the obtained Western blot data, we had mainly looked for changes in overall signaling activation, which we had then quantified as the area under the curve (AUC), and not paid too much attention on potential kinetic differences. However, we agree that the two blots depicted in the first version of the manuscript may have suggested that there is a kinetic difference in the signaling. Based on the raised question, we re-examined all blots from the other replicates of this experiment. As usually with Western blot-based analysis of signaling pathways, there is some variation between experiments. However, throughout the different experiments we could not see any consistent kinetic difference. This was objectified by summarising the densitometric data in Fig. 2h. Nevertheless, it is very obvious that in the Dectin-1a cell line the ERK activation is stronger than in the Dectin-1b line. Thus, the two blots, which were shown in the first version of the manuscript, were representative with regard to the difference in overall ERK activation. However, they were apparently not representative with regard to the kinetics of the process as they suggested a kinetic difference between Dectin-1a and Dectin-1b which is not conclusively present when all experiments are considered. Following this conclusion, we have exchanged the two blots which in light of the data summary seem to reflect the overall picture in a better way and added the quantification in Fig. 1h.

Is there evidence that the prolonged signal of the protease trimmed dimer is important? The implication is that the persistence of signaling by the cleaved receptor (that is ultimately going to be degraded and inactivated anyway) is some sort of evolved mechanism to regulate signaling via Dectin-1a differently than from Dectin-1b. The data suggest that in vitro NTF formation and SPPL2-mediated degradation isn’t particularly important for cytokine production. It presumably doesn’t regulate or affect phagocytosis since the process doesn’t begin until after internalization. It may modestly enhance or prolong ROS production, but is there any evidence that this amount of change in ROS production would be meaningful? The alternative explanation is that this is just a series of events leading to the receptor’s degradation, not a particularly relevant way by which it is regulated.

We totally agree that the increases seen in the SPPL2b-deficient cells in terms of ROS production and also cytokine secretion are rather small. When comparing our results to previous studies analyzing the loss of Cbl-b (Wirnsberger et al., *Nat. Med.* 2016; Xiao et al., *Nat. Med.* 2016; Zhu et al., *J Exp Med.* 2016), which coordinates the alternative pathway of Dectin-1 degradation, many effects reported there are in a similar range or just very slightly higher as the effects we see. As phagocytes produce ROS in order to kill and inactivate ingested pathogens, we asked whether the increased ROS production in the SPPL2b-deficient cells would translate into an enhanced killing capacity. Indeed, BMDCs and BMDMs from SPPL2b-deficient mice performed significantly better in killing *C. albicans*. The new data is depicted in Fig. 7b,c. From this

we would conclude that at least on the cellular level the increased ROS production of SPPL2b-deficient cells impacts on a pathophysiologically relevant parameter. As ROS produced by immune cells is responsible for inflammation-induced tissue damage, a regulatory loop which can adjust ROS production and by this means balance pathogen killing capacity on one hand and the risk of tissue damage on the other appears to make sense. Of course, how much impact this may have on the course of an infection *in vivo* currently remains open. Nevertheless, we feel that the new data strongly support a functional relevance.

Minor points:

The manuscript is full of novel observations. But throughout, the authors should take care to not appear to claim to first observe things others have previously reported (e.g. colocalization of Dectin-1 with LAMP, colocalization/association with Syk, etc.). I am fully in favor of reporting data that repeat what others have seen, but these data are best reported as being “consistent with previous observations”, including appropriate references.

Thanks for this suggestion. We have carefully implemented this and cited the respective original publications, where our experimental systems/approaches refer back to observations made previously by other labs. In case we may have missed something despite our serious efforts, we would be grateful for suggestions which citations still need to be added.

Supplemental 4B is noted to show that “upon inhibition with ZLL no NTF accumulation was observed”, although if the gel bands were quantified it looks like there would be some accumulation.

In the revised version the respective figure is now Suppl. Fig. 3c. We agree that upon inhibition with ZLL a very minor accumulation of a Dectin-1b-derived NTF was observed. Therefore, the previous expression in the manuscript text indeed was not adequate and has been changed to:

“ZLL treatment slightly stabilised the Dectin-1b NTF. However, the amounts were still minor when compared to Dectin-1a.”

Most importantly, this Dectin-1b NTF is produced constitutively as its abundance does not increase by treating the cells with dZym and thus by inducing receptor degradation. Therefore, the origin of this very low abundant fragment is very different from the Dectin-1a NTF and we do not have any indication that it may be involved in active signal transduction.

P. 24: “The lacking effect of siRNA against SPPL2a is in line with its failure to inhibit NTF proteolysis presumably due to knockdown efficiency at the protein level (Fig. 3I,K, Suppl. Fig. 4G).” None of these figures address whether SPPL2a protein was knocked down well or not.

In addition to the initial analysis of knockdown efficiency of SPPL2a/b by qPCR (Suppl. Fig. 3h in the updated manuscript) we now included a validation of knockdown of both proteases by Western Blotting (Suppl. Fig. 3g). This basically resembles the results from qPCR analysis demonstrating efficient, however not complete depletion of both proteases. Based on this, we cannot exclude that remaining levels of SPPL2a might impact on Dectin-1 degradation.

Is there a reason that figures 1-4 skip “J”?

This has been corrected.

REVIEWER COMMENTS

Reviewer #1 (Remarks to the Author):

In this revised version of their manuscript, Mentrup et al. have meticulously answered all of my previous concerns and questions. The manuscript has improved tremendously and I now fully support its publication in Nature Communications. Congratulations to all authors!

I only have 2 remaining minor suggestions:

1. In the results paragraph on the effects on ROS production (p14) I found it confusing that first the negative results in human cells (fig 5S) and the implications of the CD74 NTF (fig 6S) are described, and only thereafter the positive results of Fig 5 are described. I feel the manuscript might be easier to read when first Fig 5 is described and then the experiments addressing whether the same is true in human cells.

2. Please provide the figure legends of the supplementary figures with titles so the reader knows immediately what the conclusion of these figures is.

Reviewer #2 (Remarks to the Author):

I am fully satisfied with the responses to my previous queries.

Point-by-point reply:

Reviewer #1 (Remarks to the Author):

In this revised version of their manuscript, Mentrup et al. have meticulously answered all of my previous concerns and questions. The manuscript has improved tremendously and I now fully support its publication in Nature Communications. Congratulations to all authors!

Thank you very much for the positive feedback and the valuable suggestions to further improve the manuscript.

I only have 2 remaining minor suggestions:

1. In the results paragraph on the effects on ROS production (p14) I found it confusing that first the negative results in human cells (fig 5S) and the implications of the CD74 NTF (fig 6S) are described, and only thereafter the positive results of Fig 5 are described. I feel the manuscript might be easier to read when first Fig 5 is described and then the experiments addressing whether the same is true in human cells.

We have changed the order of the manuscript text as suggested. The sequence and numbering of the Supplementary Figures has been modified accordingly.

2. Please provide the figure legends of the supplementary figures with titles so the reader knows immediately what the conclusion of these figures is.

We have added titles to the supplementary figure legends as suggested.

Reviewer #2 (Remarks to the Author):

I am fully satisfied with the responses to my previous queries.

Thank you for re-evaluating the manuscript and the positive feedback